# Pretraining with hierarchical memories: separating long-tail and common knowledge

**Hadi Pouransari[1], David Grangier[2], C Thomas[2], Michael Kirchhof[2] & Oncel Tuzel[2]**
[1]Work done while at Apple    [2]Apple
pouransari@gmail.com   otuzel@apple.com

## Abstract

The impressive performance gains of modern language models currently rely on scaling parameters: larger models store more world knowledge and reason better. Yet compressing all world knowledge into parameters is unnecessary, as only a fraction is used per prompt, and impractical for edge devices with limited inference-time memory and compute. We address this shortcoming by a memory-augmented architecture and a pretraining strategy aligned with existing hardware paradigms. We introduce small language models that access large hierarchical parametric memory banks encoding world knowledge. Our pretraining learns to store long-tail world knowledge in the memory parameters, while the small language model acts as an anchor capturing common knowledge and general reasoning abilities. Through trillion-token-scale experiments, we show significant gains: a 160M-parameters model augmented with an 18M-parameters memory fetched from a 4.6B memory bank obtains comparable performance to a regular model with more than $2\times$ the parameters. We study the optimal type and size of parametric memories in transformers, scaling them to over 21B parameters. We find that our proposed hierarchical feed-forward memories work robustly across transformer architectures, whether added during pretraining or post-hoc.

## 1 Introduction

Frontier large language models (LLMs) have advanced significantly in recent years, demonstrating strong capabilities across both world knowledge and reasoning tasks. These improvements have largely been driven by scaling the number of parameters and training tokens. However, limited results exist on the role of model parameter count in each specific aspect, and on whether knowledge and reasoning can be disentangled.

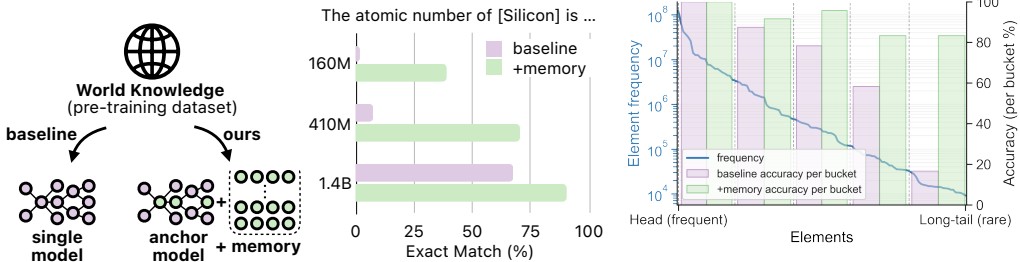

Figure 1: **Left:** Schematic of pretraining-with-memories: some parameters are always used (anchor parameters), others are fetched per input document (memory parameters). **Middle:** Accuracy improvement over baseline when $\simeq 10\%$ of parameters are allocated as memories for a knowledge-intensive task (predicting the atomic numbers of elements), using models with 160M, 410M, and 1.4B parameters, corresponding to rows A2, B2, and C2 in Table 1. **Right:** Elements sorted by their frequency of appearance in the DCLM-Baseline dataset (5 buckets, each with $\simeq 24$ elements). With the proposed *pretraining-with-memories*, we observe significant improvements, especially on long-tail data. While the baseline 1.4B model has only 17% accuracy on the least frequent element bucket, augmenting it with only 10% memory parameters increases the accuracy to 83%.

LLMs memorize many facts in their parameters during pretraining (Roberts et al., 2020), most of which are long-tail knowledge, overly specific, and unnecessary for the intended use. For example, part of the parameters of the Qwen3-2B (Yang et al., 2025) model store the fact that *Albert Einstein was born on March 14, 1879*, a detail that is not essential for executing on-device personal assistant tasks, yet is permanently loaded into RAM and considered in each computation. Ideally, this capacity would be utilized for reasoning and commonsense abilities.

We propose to use a base model as an *anchor* to capture common knowledge and reasoning capabilities and augment it with a *memory* bank that dedicates a large set of memory parameters to long-tail knowledge. In this work we focus on capturing long-tail world knowledge with parametric memories, leaving improvements to the anchor's reasoning capabilities to future work. In Fig. 1, we illustrate this through a representative knowledge-intensive task: predicting the atomic number of elements. Baseline pretraining performance degrades for samples with lower frequency in the pretraining data. This degradation is attributed to catastrophic forgetting (Toneva et al., 2018), which arises from destructive gradient updates caused by dissimilar content (Ghosal et al., 2025) applied to the same set of parameters. In contrast, in the proposed approach, memory parameters are activated and updated only on sequences of similar topics, thereby reducing susceptibility to forgetting. Besides training dynamics advantages, separating long-tail knowledge into dedicated memory parameters offers several additional benefits, as discussed below.

*Runtime efficiency*: For on-device deployment of LLMs, the main bottleneck is the availability of large and fast memory. Methods such as mixture-of-experts (Shazeer et al., 2017) (MoEs) improve compute efficiency by activating only a fraction of the feed-forward modules. Still, MoEs require on-demand random access to the full set of experts at every layer and for every token, such that all model parameters remain loaded, not just the active ones. This makes on-device deployment of MoEs challenging. In contrast to MoE models, which require refreshing the majority of their parameters (experts) at every token, in our proposed method, depending on the context, only the required knowledge parameters (about 10% of the model) are fetched and attached to the anchor model (see Fig. 2). This allows fast on-device memory to be used primarily for anchor model parameters, while knowledge parameters are stored in a slower but larger storage. Furthermore, we learn knowledge in the form of *hierarchical* memories, allowing inference to naturally align with existing device memory hierarchies (RAM, flash storage, external disk) and benefit from their compositionality over a session of interaction with the model (see Fig. 5).

*Training efficiency*: A key bottleneck in large-scale distributed training of standard LLMs is the need to communicate large gradient tensors between compute nodes. In the proposed approach, during pretraining, based on the content of documents in a batch, only a small fraction of memory bank parameters is retrieved and updated together with the anchor model parameters. As a result, the gradients are highly sparse, which substantially reduces node-to-node communication. This property makes the proposed method well-suited for co-locating training data and compute in massively distributed setups, similar to branch-train-merge (BTM) methods (Li et al., 2022; Gururangan et al., 2023a). For instance, pretraining an anchor model with 160 million parameters and 4.6 billion memory parameters requires less than $1.7\times$ the compute budget of training a 160M model alone.

*Privacy and knowledge editing*: Separating knowledge and reasoning abilities enables a direct mapping between training tokens and specific subsets of parameters (memories). Ownership of training data tokens can therefore be linked to ownership of the corresponding memories. This makes it possible to remove or edit certain data from the model by deleting or updating the associated memory parameters, or to restrict access to specific memories while keeping the anchor model public. Although knowledge editing and privacy are not the focus of this paper, we present preliminary results on the effect of memory blocking in Section 4. The proposed approach also allows creating new memories from new data after pretraining, as shown in Section 4. Through this mechanism, a strong public reasoning model can be readily augmented with memories built from large private user data or organizational databases, providing a more efficient alternative to very long contextual memories.

Our main contributions are transformer architectures augmented with large hierarchical memories (Fig. 2) and a clustering-based pretraining method (Section 2.1), through which long-tail knowledge is automatically incorporated into the memory bank. We, for the first time, systematically examine how different memory types (Fig. 3), depths (Fig. 3d), sizes (Fig. 4a), positions (Table 12), and memory-to-model ratios (Fig. 4b) affect performance. We further analyze the runtime efficiency of

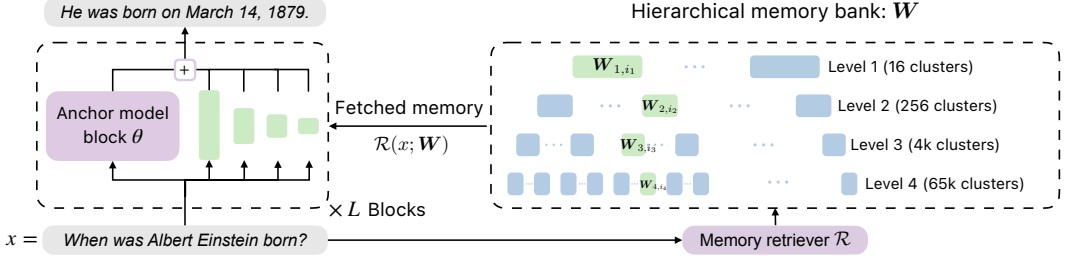

Figure 2: **Proposed architecture:** For a given context $x$ (such as a question text), the memory retriever module selects relevant parameters from a large set of memory bank parameters. These memory parameters are organized hierarchically based on the hierarchical clustering of the pretraining data (with up to $16^4 \simeq 65k$ clusters at level 4) . The anchor model, together with the retrieved memories, then responds to the question.

the proposed method in relation to hardware storage hierarchies (Section 3). Memory learning is scaled to banks containing up to 21B parameters.

In this work, we focus on demonstrating the effectiveness of the proposed parametric memories in capturing long-tail world knowledge during pretraining. The results show consistent improvements over baseline pretraining (Table 1) as well as over raw contextual memories. We further show that the proposed parametric memories can be combined with methods such as retrieval-augmented generation (Table 5) for additional gains. In Table 4, we also demonstrate robust applicability to arbitrary transformer architectures as a post-hoc augmentation. Finally, we show that with approximately 10% additional parameters retrieved from the memory bank, we observe improvements on knowledge-specific tasks from $34.1\% \rightarrow 40.3\%$ for a 160M model and from $40.9\% \rightarrow 45.9\%$ for a 410M model, confirming the runtime benefits of the proposed approach (achieving the performance of a model with more than $2\times$ the runtime parameters).

## 2 Pretraining with memories

Consider a language model with parameters $\boldsymbol{\theta}$, called anchor parameters, a large bank of memory parameters $\boldsymbol{W}$, and a retriever module $\mathcal{R}$ that, given a context $x$, fetches only the relevant memory parameters $\mathcal{R}(x; \boldsymbol{W})$ from the memory bank. During pretraining, we optimize the next token prediction loss for each document $x$ in the dataset $\mathcal{D}$:

$$\mathcal{L}(x) = -\sum_t \log \mathbb{P}_{\boldsymbol{\theta}, \mathcal{R}(x; \boldsymbol{W})}(x_t \mid x_{<t}), \tag{1}$$

where $\mathbb{P}_{\boldsymbol{\theta}, \mathcal{R}(x; \boldsymbol{W})}(x_t \mid x_{<t})$ is the model's output probability distribution over the vocabulary for token $t$ in document $x$, given its previous tokens, using anchor parameters $\boldsymbol{\theta}$ augmented with the retrieved memory parameters $\mathcal{R}(x; \boldsymbol{W})$.

In terms of parameter counts, we generally assume $|\mathcal{R}(x; \boldsymbol{W})| \ll |\boldsymbol{\theta}| \ll |\boldsymbol{W}|$. Therefore, with the training objective in Eq. (1), the memory bank parameters $\boldsymbol{W}$ are updated only sparsely. Intuitively, we expect $\mathcal{R}(x; \boldsymbol{W})$ to retrieve the same subset of parameters from $\boldsymbol{W}$ for inputs $x$ that are semantically similar. As a result, memory parameters receive gradients primarily from documents with related content, mitigating forgetting (Ghosal et al., 2025) and enabling them to efficiently *memorize* long-tail world knowledge. In contrast, $\boldsymbol{\theta}$ receives gradients for all $x \in \mathcal{D}$ and is expected to primarily capture common knowledge and general reasoning capabilities.

At inference time, the model uses only $|\boldsymbol{\theta}| + |\mathcal{R}(x; \boldsymbol{W})|$ parameters, introducing only a minor overhead compared to a model that relies solely on $\boldsymbol{\theta}$. Fig. 2 presents the overall architecture. We next describe the construction of each component in this framework.

### 2.1 Clustering-based memory retriever

**Hierarchical clustering:** Given a pretraining dataset $\mathcal{D}$, we use an off-the-shelf text embedding model $\phi$ to map each document $x \in \mathcal{D}$ to $\phi(x) \in \mathbb{R}^c$. We then perform hierarchical clustering using

the document embeddings: first, we divide the documents into $k$ clusters using $k$-means; next, we further subdivide each cluster into $k$ sub-clusters, and continue this process for $p$ levels. Finally, we obtain $k^l$ nested clusters at each level. In this paper, we cluster the DCLM-baseline dataset (Li et al., 2024) with 3.2 billion documents into a hierarchy with $p = 4$ levels and dividing factor of $k = 16$ resulting in $16, 16^2, 16^3$, and $16^4$ clusters at levels 1, 2, 3, and 4, respectively (see Fig. 2). We use Sentence-BERT `all-MiniLM-L6-v2` embedding model (Reimers & Gurevych, 2019) with dimension $c = 384$. See more details in Appendix C.

To retrieve a cluster, we map a document $x$ to an index tuple $\mathcal{I}(x) = (i_1, i_2, \ldots, i_p)$ by traversing the clustering tree greedily: at each level $l$, $\phi(x)$ is compared to $k$ centroids and the sub-tree corresponding to the closest (via L2 distance) centroid ($i_l$) is then visited. This greedy traversal leads to a fast and scalable retrieval within $\mathcal{O}(pk)$ comparisons. For our tree of depth $p = 4$, we denote the cluster index as $\mathcal{I}(x) = (i_1, i_2, i_3, i_4)$. We pre-compute the cluster index for each document in the pretraining dataset offline, resulting in no training-time overhead. See details in Appendix A.2. At test time, we use the same traversal to get the cluster index for the task context (e.g., question text).

**Hierarchical memories:** We assign a memory parameter block to each cluster in a given hierarchical clustering tree, denoted by $\boldsymbol{W}_{l,i_l} \in \mathbb{R}^{s_l}$, where $l \in \{1, 2, 3, 4\}$ denotes the level, $i_l \in \{1, \ldots, k^l\}$ is the cluster index at level $l$, and $s_l$ is the size of the memory parameter blocks at level $l$. The memory bank $\boldsymbol{W}$ consists of all memory parameter blocks $\boldsymbol{W}_{l,i_l}$. The memory retriever is then:

$$\mathcal{R}(x; \boldsymbol{W}) = [\boldsymbol{W}_{1,i_1}, \boldsymbol{W}_{2,i_2}, \boldsymbol{W}_{3,i_3}, \boldsymbol{W}_{4,i_4}], \quad \text{where } \mathcal{I}(x) = (i_1, i_2, i_3, i_4). \tag{2}$$

The total number of parameters in the memory bank is $|\boldsymbol{W}| = s_1 k + s_2 k^2 + s_3 k^3 + s_4 k^4$, and the number of retrieved parameters (*fetched memory* size) is $|\mathcal{R}(x; \boldsymbol{W})| = s_1 + s_2 + s_3 + s_4$.

## 2.2 INFERENCE WITH PARAMETRIC MEMORIES

There are multiple ways to add parametric memories $\mathcal{R}(x; \boldsymbol{W})$ to an anchor model $\boldsymbol{\theta}$, i.e., to practically model $\mathbb{P}_{\boldsymbol{\theta}, \mathcal{R}(x; \boldsymbol{W})}(x_t \mid x_{<t})$. We limit our discussion to decoder-only, transformer-based (Vaswani et al., 2017) language models. For all memory types, we instantiate them such that at the beginning of training they have no effect on the anchor model. See details in Appendix I.

**LoRa-Memories:** A popular approach to augment a model with a (small) set of extra parameters is to patch its linear layers with low-rank adaptation matrices (LoRa, Hu et al., 2022). We consider three types of LoRa memories: LoRa-QK adapts the query and key projection layers, LoRa-VO adapts the value and output projection layers, and LoRa-FFN adapts all three linear layers in the SwiGLU feed-forward network (FFN) (Shazeer, 2020) with rank $r$ matrices.

**KV-Memories:** The knowledge in the context tokens a transformer attends to is ultimately a sequence of KV-caches. A natural extension of providing context knowledge is thus to learn KV-cache parameters directly. This can also be seen as a generalization of prefix tuning (Li & Liang, 2021; Lester et al., 2021), where only input token embeddings are learned. At each transformer layer, data-dependent query tokens cross-attend to $r$ fetched KV memories.

**FFN-Memories:** Previous works have argued that transformer-based language models mainly store their knowledge in the FFN layers (Geva et al., 2020; Dai et al., 2022; Yao et al., 2022). Inspired by this, we introduce FFN memories: we expand the inner dimension of the SwiGLU FFN layers by $r$ through concatenation with the fetched memory parameters, which is equivalent to a fast addition.

The number of parameters $s_l$ in a memory block assigned to clusters at level $l$ depends on the memory type, the anchor model's architecture (e.g., hidden dimension, depth, etc.), and the memory block size multiplier $r$ (rank for LoRa, number of KV memory tokens, or FFN dimension expansion size). Therefore, a hierarchical memory configuration $(s_1, s_2, s_3, s_4)$ can be written as $c_0(r_1, r_2, r_3, r_4)$, where $c_0$ is a constant determined by the anchor model architecture and memory type, and $r_l$ is the memory block size multiplier for memories at level $l$ that we control. For simplicity, we drop the constant $c_0$ in the rest of the paper and provide details in Appendix I. In practice, we generally set these multipliers so that coarser levels have larger parameter blocks, i.e., $r_1 \geq r_2 \geq r_3 \geq r_4$, or set $r_l = 0$ when no memories are assigned to clusters at level $l$.

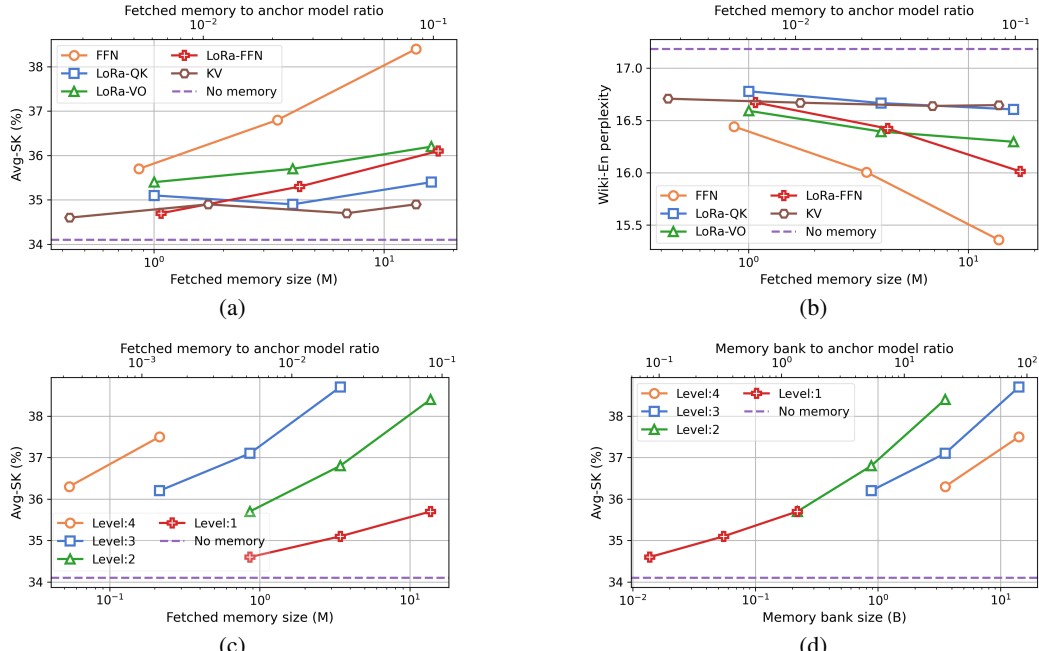

Figure 3: Effect of **memory type** on Specific-Knowledge benchmarks (a) and Wikipedia perplexity (b). Effect of **memory level** on performance as a function of fetched memory (c) and bank size (d).

# 3 DESIGN CHOICES FOR PRETRAINING WITH MEMORIES

We use DCLM-Baseline (Dai et al., 2022) for training. The dataset contains $\simeq 3.2$ billion documents ($\simeq 4.3$ trillion tokens). We cluster the dataset to a tree with 4 levels, having $16, 16^2, 16^3$, and $16^4$ clusters at each level. See Appendix A for training details. For evaluation, we consider 13 frequently used benchmarks, including multiple-choice and generative tasks. We divide these tasks into two groups based on the level of specific knowledge required (see Appendices B and G for details):

**Common-Knowledge (Avg-CK):** *Lambada-OpenAI* (Paperno et al., 2016), *BoolQ* (Clark et al., 2019), *SQuAD* (Rajpurkar et al., 2016), *Winograd* (Levesque et al., 2012), *CoQA* (Reddy et al., 2019), and *WinoGrande* (Sakaguchi et al., 2021).

**Specific-Knowledge (Avg-SK):** *Hellaswag* (Zellers et al., 2019), *ArcEasy/Challenge* (Clark et al., 2018), *TriviaQA* (Joshi et al., 2017), *NaturalQuestions-Open* (Lee et al., 2019; Kwiatkowski et al., 2019), *PIQA* (Bisk et al., 2019), and *OpenBookQA* (Mihaylov et al., 2018).

For evaluation, we retrieve the memories based only on the question text. As an open-ended generation task, we also track the average perplexity on the 2022 English Wikipedia (**Wiki-En**) with $\simeq 6.5$M samples (4B tokens), where memories are retrieved based on the full Wikipedia document.

**FFN-Memories outperform LoRa and KV memories:** In Fig. 3, we compare different memory types introduced in Section 2.2. We attach the memories to a pretrained model with 160M parameters (row A1 in Table 1) and train them from scratch for 275B tokens with the loss objective in Eq. (1), while the anchor model parameters ($\boldsymbol{\theta}$) are frozen. For this set of experiments, we use a single-level memory configuration in the form of $(0, s_2, 0, 0)$, corresponding to a total of $16^2 = 256$ memories, each with size $s_2$, for different values of $s_2$. We observe that FFN-Memories have a significant advantage over other forms of memory across all memory sizes. Based on this observation, in the rest of the paper we use only FFN-Memories.

**Accuracy improves with deeper and larger memories:** Here, we explore the design space of different single-level memory configurations $(s_1, s_2, s_3, s_4)$, where only one of the $s_l$'s is non-zero. The anchor model is pretrained and frozen (row A1 in Table 1) during memory training (see Appendix A.3 for details). As shown in Fig. 3c, for a constant fetched memory size (i.e., $s_l$), deeper memories yield greater accuracy improvements, as they capture more relevant and detailed information for a given query. In Fig. 3d, we show that performance is a strictly increasing function of total

memory bank size for all memory configurations, as more capacity becomes available to capture long-tail knowledge. Shao et al. (2024) recently made a similar observation for regular RAG setups by increasing the size of the datastore from which documents are retrieved. Note that for a fixed total memory bank size, a shallower memory corresponds to a larger fetched memory size. Therefore, in Fig. 3d, at a fixed memory bank size, shallower memories achieve higher accuracy.

**Hierarchical memories enable an optimal design:** Unlike single-level memories used in previous experiments, a general hierarchical memory configuration $(s_1, s_2, s_3, s_4)$ with possibly multiple non-zero values allows independent control of the total memory bank size $(\sum_l 16^l s_l)$ and the size of fetched memory parameters at inference time $(\sum_l s_l)$. For a large total memory bank size with a small number of fetched parameters at inference, we can use larger level-3 and level-4 memories. Conversely, for a smaller total bank size with more fetched parameters at inference, we can increase the sizes of level-1 and level-2 memories.

From a learning dynamics perspective, in regular language modeling, all parameters are updated at every iteration, receiving gradients from a wide range of dissimilar documents in the dataset. As a result, long-tail information is often forgotten (Toneva et al., 2018). When training with hierarchical memories, however, memory bank parameters at level $l$ are activated $16^l$ times less frequently compared to anchor model parameters (which can be considered level-0). Consequently, deeper memory bank parameters receive fewer gradient updates, and those updates come from more similar content, shielding them from forgetting (Ghosal et al., 2025) and enabling them to learn long-tail knowledge. In contrast, shallow memories are updated by many more occurrences of common facts in the dataset, giving commonsense knowledge a greater chance to dominate their gradients. This leads to effective learning of a hierarchy of memories, that intuitively range from the most common knowledge at level 1 to the most specific knowledge at level 4.

Above, we showed that single-level memories benefit from larger fetch size and bank size. We now systematically demonstrate this for a general hierarchical configuration. We evaluate two groups of hierarchical memories added to our frozen pretrained 160M model (row A1 in Table 1): one with a memory bank size of 4.6B and another with 18.7B, both with configurations spanning between 1M and 300M fetched parameters (see Appendix A.4 for details). Results in Fig. 4a confirm that performance increases strictly with fetched memory size (while keeping bank size fixed) and with bank size (while keeping fetched memory size fixed) for the general hierarchical configurations.

The point with highest accuracy in Fig. 4a corresponds to augmenting a 160M anchor model with $\simeq 240$M fetched memory parameters, achieving 44.5% accuracy on Avg-SK. This is a notable result, showing that an anchor+memory model with a total of 400M runtime parameters outperforms a regularly trained 410M model (row B1 in Table 1) by 3.6 points. Building on this observation, we ask the following question: *For a given runtime parameters budget, how many parameters should be allocated to the anchor model and how many to the fetched memories?*

To explore this, we train a sequence of anchor models with 260M, 320M, 350M, 370M, 385M, and 410M parameters, freeze them, and pair them with fetched memories of sizes 150M, 90M, 60M, 40M, 25M, and 0, respectively, such that the total runtime parameter count for anchor+memory remains fixed at 410M. All configurations use a 6.3B memory bank (see Appendix A.7). In Fig. 4b, a 1:10 ratio of fetched memory to anchor model size appears optimal and guides our next experiments. However, this observation may not generalize to different runtime, memory bank, and training budgets, or when the anchor model and memory parameters are co-trained.

**Models with hierarchical memories have on-device deployment advantages.** Assuming a von Neumann architecture with hardware organized in increasing size and decreasing speed (RAM, flash disk, external disk), we can store shallower memory levels (with larger fetch size but smaller bank size) on faster components, while offloading deeper levels to slower storage (Alizadeh et al., 2024). In the example in Fig. 5 (left), with **a hypothetical hardware setup**, a hierarchical memory can be loaded in 38ms, whereas loading a memory of the same fetch size from a flat bank (stored on the external disk due to its excessive total size) takes 198ms—more than $5\times$ longer. Additionally, even if both hierarchical and flat memory banks are stored exclusively on the external disk, hierarchical memories still provide a loading speed advantage due to their *compositionality*. For example, when generating different atomic numbers in Fig. 1, the level-1 and level-2 memories remain mostly unchanged, and only deeper memories need to be swapped. This takes 47ms, compared to 198ms when the entire flat memory must be reloaded, as shown in Fig. 5 (right).

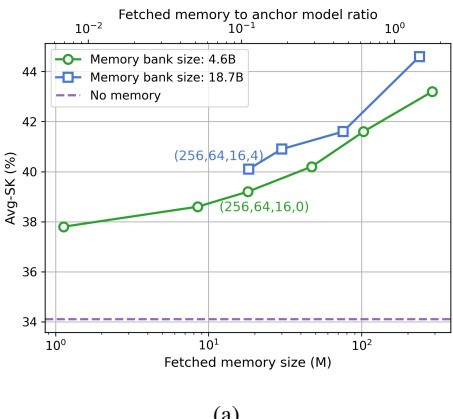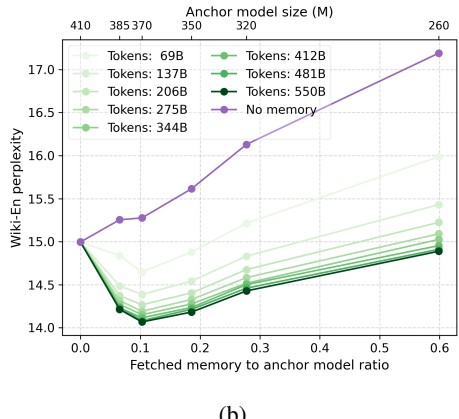

(a)                                                                        (b)

Figure 4: (a) Avg-SK accuracy for different hierarchical memories, demonstrating **performance gain with larger bank size and fetched memory size**. (b) Wiki-En perplexity for different **fetched memory–to–anchor model size ratios, with the optimal point at 1:10**. The purple curve shows the perplexity of anchor models without memory. The green curves show the perplexity of models with memory, with different shades of green corresponding to the progress of memory training.

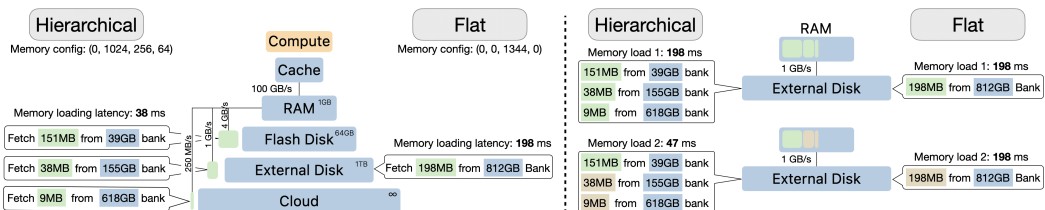

Figure 5: Deployment advantages of hierarchical memories. **Left:** Memory loading latency is reduced by using the hardware hierarchy. **Right:** Latency is reduced by exploiting compositionality over time: larger memories for low-level clusters, once loaded, are less likely to need reloading.

**Sensitivity to data clustering.** In Table 2 we report sensitivity to the initial data clustering, fixing the 160M anchor model frozen and a memory configuration with 4.6M fetch size and 4.7B memory bank size, so clustering is the only difference between experiments. To assess sensitivity to the embedding model, we replace the `all-MiniLM-L6-v2` embedding model (Reimers & Gurevych, 2019) with `GTE-Small` Li et al. (2023), which Grangier et al. (2025) show has better clustering performance; nonetheless, we observe very small sensitivity in the final model performance. We further study the choice of clustering depth $p$ and dividing factor $k$ by considering $p = 2$ and $k = 256$, which yields slight improvements on knowledge-specific tasks (Table 2). Note that two-level nested clustering with dividing factor 16 leads to a suboptimal one-level clustering with dividing factor 256, consistent with this observation. In practice, the optimal choice of clustering and memory configuration also depends on the underlying hardware hierarchy used to store and fetch the memory bank.

# 4    RESULTS

In this section, building on the findings from Section 3, we provide a comprehensive set of results for different-sized models and compare them with baselines. See Appendix A for training details.

Starting from a 160M anchor model pretrained regularly (row A1 in Table 1), we add memories with the $(256, 64, 16, 0)$ configuration, corresponding to a fetched memory size of $\simeq 18$M parameters and a total memory bank size of $\simeq 4.6$B parameters. When memories are learned with a frozen anchor model (row A3), we observe a +5.1 points improvement on Avg-SK compared to A1 and a 2 points reduction in Wiki-En perplexity, demonstrating the effectiveness of memories for tasks requiring specific knowledge, with only $\simeq 10\%$ additional runtime parameters from fetched memories.

To ensure a fair comparison, we also train a *generic* memory with the same size as the fetched memories (18M parameters) together with the memory bank parameters. When evaluating with

generic memory, unlike fetched memories that are retrieved based on context, we simply use the anchor+generic memory parameters. This isolates the effect of merely increasing the number of parameters and training tokens in the anchor model. Anchor+generic memory scores 34.7% on Avg-SK, 4.5 points below fetched memories, showing the benefit of context-based retrieval in isolation.

We next explore co-training the anchor model parameters ($\boldsymbol{\theta}$) and the memory parameters ($\boldsymbol{W}$) together, as in Eq. (1). For a fair comparison, during training we use the generic memory with probability $1/(16+1)$ and the fetched memory with probability $16/(16+1)$, where 16 is the clustering division factor. This ensures there is no training bias in favor of the memory bank parameters.

Row A2 in Table 1 shows co-training results, with Fig. 6a illustrating training curves for A2 and A3 experiments. A key observation is that when we allow the anchor parameters to be co-trained with the memory bank, we obtain greater improvement compared to the case where the anchor model is frozen (Avg-SK $39.2\% \rightarrow 40.3\%$). This gain can be attributed to two factors: 1) when co-training, the anchor model learns to utilize the memories more effectively compared to when it is frozen, and 2) the anchor model performance improves simply due to more training. The latter effect should be minor if the anchor model is already converged. We provide additional discussion in Appendix E.

We also explore co-training the anchor model and memory bank together *from scratch*. Results are shown in row A4 of Table 1. Despite using the same total training budget as row A2, A4 shows lower performance. This result suggests that memories are learned more effectively after the anchor model has been trained to some extent (i.e., the setup of row A2). This is analogous to human memory, which develops only after the brain gains semantic understanding, around age 3 (Shaw, 2016).

Next, we scale the co-training setup of row A2 to anchor models of sizes 410M and 1.4B, corresponding to rows B2 and C2 in Table 1. These models are paired with memory configurations such that the fetched memory is approximately 10% of the anchor model size, corresponding to memory bank sizes of 12.7B and 21.1B for the 410M and 1.4B anchor models, respectively. We observe similar improvements for these larger models: for Avg-SK, fetched memories outperform generic memories by +4.1 points for the 410M and +3.6 points for the 1.4B model.

Finally, for the 410M model, we train a model regularly (without memories) with the same total training budget as row B2 (2.2T tokens). We observe that Avg-CK performance is worse than that of the anchor+generic setup in row B2. These preliminary results suggest that when the anchor model is co-trained with a large memory bank (as in row B2), long-tail knowledge is offloaded to the memories, enabling the anchor model to perform better on common-knowledge tasks.

**Inference cost:** In Table 3 we show inference time (ms) for the 1.4B anchor model without and with memory (rows C1 and C2 in Table 1) on a H100 NVIDIA GPU. We consider two setups: (1) memory bank on GPU High Bandwidth Memory (HBM) and (2) memory bank on CPU DRAM. Routing runs on CPU once per generation and adds only minor overhead. With all overheads included (routing, fetch, and extra inference flops), the total time increases by less than 10% for a 40-token generation.

**Blocking part of the memory bank:** In Fig. 6b, we show the performance of the 410M model with memories (row B2 in Table 1) on predicting atomic numbers of elements (see Fig. 1) when the best-matching parts of the memory bank are adversarially blocked during retrieval. We observe a significant performance drop, from 70% to 20%, when blocking 1/16 of the bank. This preliminary result highlights the potential of the proposed approach for applications with privacy goals.

**Failure analysis of retriever:** In Appendix J we present analysis of model performance when wrong memories are fetched. We observe that even when fetched memories are not the nearest neighbors in the clustering, overall performance is on par with or better than the no-memory baseline.

**Adding memory to other pretrained models:** We explore the post-hoc addition of parametric memories to open-weight models. We span multiple sizes and architectures, namely Gemma 3 270M (Gemma Team et al., 2025), Qwen 2.5 0.5B (Yang et al., 2024), and Llama 3.2 1B (Meta AI, 2024). We add hierarchical FFN memories of $\simeq 10\%$ the model size post-hoc to pretrained (and frozen) anchor models and train the memories for 1.1T tokens on DCLM, see Appendix A.6 for additional details. Results are shown in Table 4. As above, the specific knowledge accuracy improves with memories. This is consistent across all architectures, showing the generality of adding hierarchical memories across all models. Common knowledge remains the same or slightly decreases; potentially

| Row | Anchor model | Init. | Seen tokens | Cotrain anchor | Memory config | Bank size | Fetch size | Avg-CK (%) ↑ | | Avg-SK (%) ↑ | | WikiEn Pplx ↓ | |
|---|---|---|---|---|---|---|---|---|---|---|---|---|---|
| | | | | | | | | Generic | Fetched | Generic | Fetched | Generic | Fetched |
| A1 | | Scratch | 1.1T | n/a | (0,0,0,0) | 0 | 0 | 45.9 | | 34.1 | | 17.2 | |
| A2 | 160M | A1 | +1.1T | Yes | | | | 47.9 | **48.7** | 35.7 | **40.3** | 16.7 | **14.2** |
| A3 | | A1 | +1.1T | No | (256,64,16,0) | 4.6B | 18M | 46.6 | 47.4 | 34.7 | 39.2 | 16.7 | 15.2 |
| A4 | | Scratch | 2.2T | Yes | | | | 46.6 | 46.7 | 33.8 | 39.6 | 17.8 | 15.6 |
| B1 | | Scratch | 1.1T | n/a | (0,0,0,0) | 0 | 0 | 52.3 | | 40.9 | | 13.9 | |
| B2 | 410M | B1 | +1.1T | Yes | (512,128,32,0) | 12.7B | 50M | 55.5 | **56.1** | 41.8 | **45.9** | 13.8 | **12.4** |
| B3 | | Scratch | 2.2T | n/a | (0,0,0,0) | 0 | 0 | 53.2 | | 41.1 | | 13.8 | |
| C1 | 1.4B | Scratch | 1.1T | n/a | (0,0,0,0) | 0 | 0 | 61.2 | | 49.7 | | 10.8 | |
| C2 | | C1 | +1.1T | Yes | (768,256,16,0) | 21.1B | 153M | 64.4 | **64.5** | 51.3 | **54.9** | 11.0 | **10.2** |

Table 1: Results for pretraining with memories at different scales. See full results in Table 11.

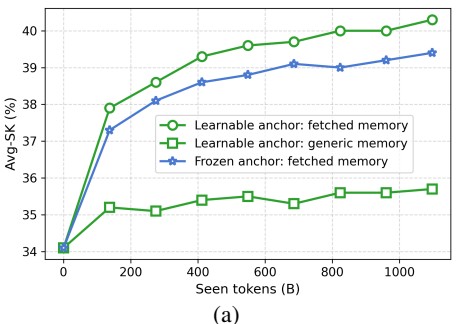
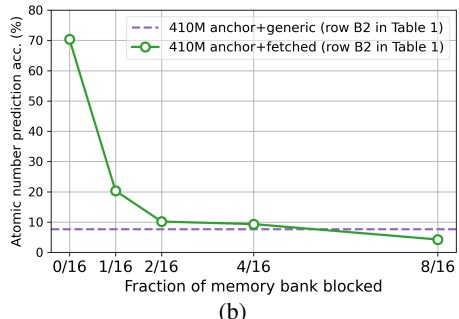

Figure 6: (a) Performance improvements on Avg-SK when co-training memories and anchor model parameters jointly during training. (b) Effect of blocking parts of the memory bank from retrieval.

because these models were pretrained on more tuned data mixtures than the simple open-source data mixture DCLM that we use in these experiments.

**Comparing with vanilla retrieval-augmented generation:** An alternative, yet complementary, approach to parametric memory is retrieval-augmented generation (RAG), where relevant texts are retrieved from a datastore and prepended to the context (Lewis et al., 2020; Ram et al., 2023; Izacard et al., 2023) to improve performance on knowledge-intensive tasks. To study the two memory mechanisms, contextual memories as in RAG and parametric memories as in the proposed approach, in a *controlled experiment*, we use our Sentence-BERT embedding model as the retriever and the DCLM training data as the datastore, making the memory mechanism the only difference. We then evaluate RAG on the baseline models from rows A1, B1, and C1. See implementation details in Appendix H.

As shown in Table 5, vanilla retrieval from DCLM performs poorly relative to the baseline models, while adding more than $2\times$ runtime FLOPs and requiring large storage for the raw-document datastore. This is likely due to the low quality of DCLM (a pretraining dataset) when used as a RAG datastore. To give RAG an advantage, we also retrieve from the higher-quality Wiki-En. RAG-Wiki improves baseline performance on SK (e.g., from 46.9 to 49.2 for the 1.4B model) while remaining slightly below baseline on CK. By contrast, learned memories (with $\simeq 10\%$ extra parameters) improve both CK and SK, with lower FLOPs overhead. We note that a more sophisticated RAG system is possible for instruction-following models after an appropriate post-training stage, and is complementary to the proposed pretraining learned-memory approach. The two can be combined for additional gains. As a demonstration of this complementarity, combining RAG-Wiki with 10% parametric memories for the 410M model in Table 5 boosts AVG-SK performance to 45.7%, surpassing either RAG-Wiki (41.6%) or the proposed parametric memories (44.5%) alone.

## 5 RELATED WORKS

**Databases.** (Ahn et al., 2016) use a symbolic knowledge graph, and Borgeaud et al. (2022) introduce *Retro*, augmenting language-model predictions with a large raw-text memory bank. More recently, Zhao et al. (2025) propose replacing long-tail knowledge with retrieval to an external knowledge base by masking retrieved tokens during pretraining. Limitations of these approaches are scalability to large pretraining corpora and low compression rate because the memory stores raw text.

**Parametric memories.** Wu et al. (2022) propose memorizing transformers, which use nearest-neighbor lookup to sparsely retrieve cached key–value pairs. For efficiency, Eyuboglu et al. (2025)

| Cluster config. | Embed. model | Memory config. | AVG-CK (%) | AVG-SK (%) |
|---|---|---|---|---|
| $p = 4$ $k = 16$ | MiniLM V6 | (64,16,4,1) | 42.8 | 36.6 |
| $p = 4$ $k = 16$ | GTE small | (64,16,4,1) | 42.6 | 36.2 |
| $p = 2$ $k = 256$ | MiniLM V6 | (84,1) | 43.1 | 37.5 |

Table 2: Clustering ablation.

| Num tokens | Model setup | Bank Loc. | Routing time | Fetch time | Prefill and Generation time | Total time |
|---|---|---|---|---|---|---|
| 20 | Baseline | n/a | n/a | n/a | 507 ms | 507 ms |
| 20 | w/ mem. | HBM | 7 ms | 0 | 534 ms | 541 ms |
| 20 | w/ mem. | DRAM | 7 ms | 23 ms | 534 ms | 564 ms |
| 40 | Baseline | n/a | n/a | n/a | 1017 ms | 1017 ms |
| 40 | w/ mem. | HBM | 7 ms | 0 | 1040 ms | 1047 ms |
| 40 | w/ mem. | DRAM | 7 ms | 23 ms | 1040 ms | 1070 ms |

Table 3: Runtime cost comparison on a single H100 GPU.

| Pretrained model | Bank size | Fetch size | Avg-CK (%) ↑ | Avg-SK (%) ↑ | Atomic Number Acc. (%) ↑ |
|---|---|---|---|---|---|
| Gemma 3 270M | 0 | 0 | 44.2 | 34.3 | 1.7 |
| + memory | 5.9B | 23.2M | **44.8** | **38.2** | **49.2** |
| Qwen 2.5 0.5B | 0 | 0 | **53.9** | 40.6 | 53.4 |
| + memory | 11.1B | 43.4M | 52.1 | **44.5** | **90.4** |
| Llama 3.2 1B | 0 | 0 | **58.9** | 46.6 | **96.6** |
| + memory | 14.1B | 102.2M | 57.6 | **50.5** | **96.6** |

Table 4: Learning memory on top of pretrained open-weight models post-hoc. All trainings use 1.1 trillion tokens from DCLM.

| Anchor model | Inference setup | Bank size | FLOPs | Avg-CK 0-shot | Avg-SK 0-shot |
|---|---|---|---|---|---|
| 160M | Baseline | 0 | ×1 | 43.6 | 32.8 |
| | RAG-DCLM | 70 TB | ×2.3 | 42.4 | 32.6 |
| | RAG-Wiki | 21 GB | ×1.7 | 42.4 | 35.0 |
| | 10% Memory | 9 GB | ×1.11 | **45.3** | **38.4** |
| 410M | Baseline | 0 | ×1 | 48.6 | 38.5 |
| | RAG-DCLM | 70 TB | ×2.3 | 48.2 | 38.1 |
| | RAG-Wiki | 21 GB | ×1.7 | 47.3 | 41.6 |
| | 10% Memory | 25 GB | ×1.1 | **52.0** | **44.5** |
| 1.4B | Baseline | 0 | ×1 | 56.0 | 46.9 |
| | RAG-DCLM | 70 TB | ×2.3 | 55.8 | 46.1 |
| | RAG-Wiki | 21 GB | ×1.7 | 55.5 | 49.2 |
| | 10% Memory | 42 GB | ×1.1 | **59.3** | **52.4** |

Table 5: Comparison with vanilla RAG.

introduce *Cartridges*: KV-memories (similar to what we discussed in Section 2.2) that learn a specific long document as a more runtime-efficient alternative to in-context learning. We find that KV-memories underperform compared to FFN-memories, at least for large-scale memorization. Mem-Sinks (Ghosal et al., 2025) uses a type of FFN-memories, where they dedicate a fraction (e.g., 30%) of FFN neurons per layer to memorization. However, their goal is to throw those parameters away at inference time for privacy and generalization. Our context-dependent memory retrieval is also conceptually related to instruction-following pruning Hou et al. (2025), which selects the most suitable parameters from a larger model based on the task description.

**Mixture of experts.** Our approach is related to MoEs (Shazeer et al., 2017). Jelassi et al. (2024) show that increasing the number of experts improves memorization while reasoning saturates when active parameters are fixed, aligning with our anchor-memory decomposition to balance reasoning and knowledge. For privacy, Shi et al. (2025) propose *FlexOlmo*, combining a publicly trained anchor expert with exchangeable domain experts trained on private data. Product key memory (PKM) (Lample et al., 2019) can be seen as a sparser MoE that combines two selected expert sets; Huang et al. (2024) improve PKM via tensor decomposition, and Berges et al. (2024) augment subsets of FFN layers with such memory to boost factual benchmarks. These MoE approaches are similar in kind to ours, and we expect that some of our insights may carry over. However, our memory architecture is vastly different, allowing to offload inactive parameters based on the context, give explicit control over memories during both training and inference, and add memories post-hoc.

# 6 DISCUSSION

**Conclusion.** We propose pretraining language models with memories to automatically capture long-tail world knowledge in large hierarchical memory banks. Small language models augmented with memory banks match regular transformers with $2\times$ more parameters. Moreover, we propose and analyze several additional potential benefits of this design, including more efficient hardware implementation and enhanced data privacy.

**Limitations and future directions.** One unexplored direction is the study of optimal scaling law formulas for learning memories. Compute-optimal scaling laws developed for dense training (Hoffmann et al., 2022), where training tokens can potentially update all learnable parameters, are not necessarily applicable to pretraining with memories, since memory parameters are updated less frequently and receive gradients only from a subset of the dataset. We also mainly focused on architecture design for memories, and leave architecture search and design of anchor models, including dynamic and learned retriever mechanisms, as promising future work. Another future direction is to explore methods to improve the anchor model's reasoning capabilities when co-trained with memories. Finally, pretraining with memories can benefit multilingual setups (not considered in this work) or other modalities beyond text, such as vision. We leave this investigation for future work.

## REPRODUCIBILITY STATEMENT

Our implementation is based on the publicly available repository OpenLM (Gururangan et al., 2023b), which we will release after the review period. In particular, our proposed memories can be added to arbitrary open-weight models and trained on any data of interest. We expect that releasing this code will enable other researchers to further investigate this direction. To ensure full transparency of our setup, especially during the review period, we provide detailed implementation descriptions in the supplementary material, including training details in Appendix A, evaluation setup in Appendix B, and clustering details in Appendix C.

## ACKNOWLEDGMENTS

We would like to thank Cheng-Yu Hsieh, Samira Abnar, Stephen Pulman, Karen Khatamifard, Chun-Liang Li, and Rick Chang for their feedback.

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

# A    TRAINING DETAILS

In this section, we provide the training details of all experiments. All experiments in the paper are conducted using the OpenLM (Gururangan et al., 2023b) repository[1]. The training data is DCLM-Baseline (Li et al., 2024). Rotary Positional Embedding (RoPE) (Su et al., 2024) is used to encode positions in queries and keys before the attention module, with a frequency of $100,000$. For all training jobs, anchor and memory parameters are stored in BFloat16 precision and trained with PyTorch Fully Sharded Data Parallelism (FSDP) using AdamW optimizer ($\beta_1 = 0.9, \beta_2 = 0.95$) and gradient clipping to 1.0. We use NVIDIA H100 GPUs for all experiments in this paper.

## A.1    BASELINE MODEL TRAINING

The baseline models, corresponding to rows A1, B1, and C2 in Table 1, are transformers with the block shown in Fig. 13 and configurations detailed in Table 6, Table 7, and Table 8, respectively. Each model is trained from scratch with $2^{40} \simeq 1.1$T tokens, using a context length of 8,192 and the dataset decomposition approach (Pouransari et al., 2025) with a global batch size of 1M tokens. The learning rate follows a cosine schedule with 10k warmup steps, a maximum value of $5 \times 10^{-3}$, and a cooldown value of $5 \times 10^{-5}$. Weight decay is set to 0.05.

| Model | OpenLM-160M |
|---|---|
| **Num layers** | 35 |
| **Hidden dim** | 512 |
| **Num heads** | 12 |
| **Per head dim** | 32 |
| **FFN inner dim** | 2,048 |
| **Head-Embedding** | tied |
| **Num params** | 163,510,016 |

Table 6: OpenLM-160M

| Model | OpenLM-410M |
|---|---|
| **Num layers** | 24 |
| **Hidden dim** | 1,024 |
| **Num heads** | 16 |
| **Per head dim** | 64 |
| **FFN inner dim** | 2,816 |
| **Head-Embedding** | separate |
| **Num params** | 411,665,408 |

Table 7: OpenLM-410M

| Model | OpenLM-1B |
|---|---|
| **Numb layers** | 24 |
| **Hidden dimension** | 2,048 |
| **Num heads** | 16 |
| **Per head dim** | 128 |
| **FFN inner dim** | 5,632 |
| **Head-Embedding** | separate |
| **Num params** | 1,439,893,504 |

Table 8: OpenLM-1B

## A.2    MEMORY MODEL TRAINING

For all memory learning jobs, we first perform clustering as described in Appendix C. After identifying the corresponding level 4 cluster ID of each document (a number between 1 and $16^4$), we pack documents from the same cluster into sequences of length 2,048, globally shuffle them, and add the cluster ID as a prefix to each sequence. During training, the cluster ID is simply obtained by separating the first token from the rest of the tokens, which represent the actual data. Note that shallower-level cluster IDs can be inferred from the level-4 cluster ID due to the nested structure of our clustering. Each sequence of length 2,048 can contain subsequences from different documents (within the same cluster), separated by a special EOT token. The attention mask is restricted to each document to avoid cross-document attention. The global batch size for all jobs is 2M tokens (1,024 sequences of length 2,048 each), except for the 1.4B model (row C2 in Table 1), where we use a global batch size of 4M to improve GPU utilization.

When the anchor model is frozen, we train memories asynchronously (one job for the memories of each level-1 subtree, resulting in a total of 16 jobs) and merge the checkpoints afterward.

In addition, we use a cosine learning rate schedule with a maximum value of $10^{-4}$ and a cool-down value of $10^{-5}$. When training for $2^{40}$ tokens, we use 10k warmup steps, and for all other trainings with different numbers of tokens, we keep the same warmup ratio. We found that warmup has minimal effect on the performance of memory learning. We use a weight decay value of $10^{-3}$, which we found to improve the performance of memory learning, consistent with its goal of memorization.

## A.3    ADDITIONAL DETAILS OF MEMORY TYPE AND SIZE ABLATIONS

The total number of seen tokens for the memory type and size experiments in Fig. 3 is $2^{38}$. For all types of memories considered, the memories correspond to level-2 clusters.

---

[1]https://github.com/mlfoundations/open_lm

For the experiments in Fig. 3d, the total seen tokens are $2^{36}$, $2^{38}$, $2^{40}$, and $2^{41}$ for memories at levels 1, 2, 3, and 4, respectively. Since deeper memories are updated less frequently, we increased the total seen token budget for those cases.

## A.4 ADDITIONAL DETAILS OF ABLATION ON MEMORY CONFIGURATION

For the experiments in Fig. 4 (corresponding to Table 9), we use $2^{40}$ and $2^{41}$ total seen tokens, corresponding to memory bank sizes of 4.6B and 18.7B, respectively.

| Memory config | | | | Bank size | Fetched size (M) | Common-Knowledge (%) | Specific-Knowledge (%) |
|---|---|---|---|---|---|---|---|
| level 1 | level 2 | level 3 | level 4 | | | | |
| 0 | 0 | 0 | 0 | 0 | 0 | 45.9 | 34.1 |
| 0 | 16 | 4 | 1 | | 1.1 | 46.5 | 37.8 |
| 98 | 42 | 18 | 0 | | 8.5 | 46.5 | 38.6 |
| 256 | 64 | 16 | 0 | 4.6B | 18.1 | 47.4 | 39.2 |
| 768 | 96 | 12 | 0 | | 47.1 | 47.6 | 40.2 |
| 1792 | 112 | 7 | 0 | | 102.7 | 48.3 | 41.6 |
| 5376 | 0 | 0 | 0 | | 289 | 49.7 | 43.2 |
| 256 | 64 | 16 | 4 | | 18.1 | 47.2 | 40.1 |
| 328 | 156 | 74 | 0 | 18.7B | 30 | 46.3 | 40.9 |
| 1216 | 164 | 22 | 3 | | 75.5 | 47.8 | 41.6 |
| 4096 | 320 | 17 | 2 | | 238.4 | 49.2 | 44.6 |

Table 9: Ablation on different hierarchical memory configurations corresponding to results shown in Fig. 4a. For all experiments, the anchor model is the 160M model (row A1 in Table 1), frozen during memory learning. Memories are trained for 1.1T and 2.2T tokens when the memory bank size is 4.6B and 18.7B, respectively.

## A.5 ADDITIONAL DETAILS OF CO-TRAINING MEMORIES AND ANCHOR MODEL

For the experiments in Table 1, we denote the total number of seen tokens, which is either $2^{40}$ or $2^{41}$. For the A4 model in Table 1, where both memory and anchor model parameters are trained together from scratch, we use a cosine learning rate schedule with a maximum value of $10^{-3}$ and a cool-down value of $10^{-5}$, with 10k warmup steps and weight decay set to 0.1.

## A.6 ADDITIONAL DETAILS OF MEMORY LEARNING ON OPEN-WEIGHT MODELS

For all experiments in Table 4, we use a total of $2^{40}$ tokens. All models are initialized from their public pretrained checkpoints, and these anchors remain frozen. We use a memory config of (512, 128, 32, 0) for Gemma and Qwen (which results in higher memory parameters for Qwen because it has a higher internal dimension and more transformer blocks) and (768, 256, 32, 0) for Llama. These numbers result in a roughly 10% memory to anchor model parameter ratio. We follow the optimizer setup of Appendix A.2, except that we do not restrict cross-attention because the baseline models were not all trained this way and reduce warmup steps to 5k.

## A.7 ADDITIONAL DETAILS OF MEMORY-TO-MODEL SIZE EXPERIMENT

Here we provide implementation details for the experiments in Fig. 4b.

We summarize the architecture of six anchor models and their corresponding memories for this experiment in Table 10. The anchor models are first trained for $2^{38}$ tokens, then frozen, and their memories are trained for another $2^{39}$ tokens. For the anchor models, we use a cosine learning rate schedule with a maximum learning rate of $10^{-3}$ and a cool-down value of $10^{-5}$. Weight decay is set to 0.1. We then freeze the anchor model to train the memories. For this particular experiment, memories are trained with a constant learning rate of $10^{-4}$ to demonstrate convergence of memory parameters, even without learning rate annealing.

| Anchor model | | | Memory | | | Runtime params |
|---|---|---|---|---|---|---|
| **Name** | **Num layers** | **Num params** | **Config** | **Fetch size** | **Bank size** | |
| 260M | 12 | 257,475,584 | `(3840,336,6,0)` | 154,165,248 | 6,341,787,648 | 411,640,832 |
| 320M | 17 | 321,721,344 | `(1445,256,8,0)` | 89,250,816 | 6,341,246,976 | 410,972,160 |
| 350M | 19 | 347,419,648 | `(870,226,9,0)` | 64,496,640 | 6,341,099,520 | 411,916,288 |
| 370M | 21 | 373,117,952 | `(384,200,10,0)` | 38,320,128 | 6,341,787,648 | 411,438,080 |
| 385M | 22 | 385,967,104 | `(264,94,16,0)` | 25,276,416 | 6,341,001,216 | 411,243,520 |
| 410M | 24 | 411,665,408 | `(0,0,0,0)` | 0 | 0 | 411,665,408 |

Table 10: Anchor models and their corresponding memory configurations for the memory-to-model size experiment in Fig. 4b. The anchor model architecture and memory configuration are designed to keep the total runtime number of parameters and the memory bank size (almost) fixed. The anchor model architecture uses a hidden dimension of 1,024 and 16 heads for all rows.

# B  EVALUATION

We use LLM-Foundry (V0.9) as the evaluation framework. To obtain a stable signal, we only include benchmarks for which the baseline 410M model (B1 in Table 1) performs better than random and the ratio of standard deviation to mean performance is less than 0.5. Below is a detailed description of the 13 tasks we use to evaluate pretrained language models in this work:

- Lambada-OpenAI (Paperno et al., 2016) with 5,153 samples, evaluated 0-shot, is of type `language modeling`, with a random baseline performance equal to 0%.
- BoolQ (Clark et al., 2019) with 3,270 samples, evaluated 0-shot, is of type `multiple choice`, with a random baseline performance equal to 50%.
- SQuAD (Rajpurkar et al., 2016) with 10,570 samples, evaluated 3-shots, is of type `language modeling`, with a random baseline performance equal to 0%.
- Winograd (Levesque et al., 2012) with 273 samples, evaluated 3-shots, is of type `schema`, with a random baseline performance equal to 50%.
- WinoGrande (Sakaguchi et al., 2021) with 1,267 samples, evaluated 5-shots, is of type `schema`, with a random baseline performance equal to 50%.
- CoQA (Reddy et al., 2019) with 7,983 samples, evaluated 0-shot, is of type `language modeling`, with a random baseline performance equal to 0%.
- Hellaswag (Zellers et al., 2019) with 10,042 samples, evaluated 0-shot, is of type `multiple choice`, with a random baseline performance equal to 25%.
- ArcEasy (Clark et al., 2018) with 2,376 samples, evaluated 3-shots, is of type `multiple choice`, with a random baseline performance equal to 25%.
- ArcChallenge (Clark et al., 2018) with 1,172 samples, evaluated 3-shots, is of type `multiple choice`, with a random baseline performance equal to 25%.
- TriviaQA (Joshi et al., 2017) with 3,000 samples, evaluated 3-shots, is of type `generation task with answers`, with a random baseline performance equal to 0%.
- PIQA (Bisk et al., 2019) with 1,838 samples, evaluated 0-shot, is of type `multiple choice`, with a random baseline performance equal to 50%.
- OpenBookQA (Mihaylov et al., 2018) with 500 samples, evaluated 10-shots, is of type `multiple choice`, with a random baseline performance equal to 25%.
- NaturalQuestions (Lee et al., 2019; Kwiatkowski et al., 2019) with 2,655 samples used by Liu et al. (2024), evaluated 0-shot, is of type `generation task with answers`, with a random baseline performance equal to 0%.

In Appendix G, we present our analysis dividing the above tasks into two groups: common-knowledge (CK) and specific-knowledge (SK).

When evaluating models with fetched memory, we retrieve memory based on the *question* for multiple choice tasks, the *context* for language modeling tasks, and the portion of the text that is common across all choices for the schema tasks. To compute perplexity (for the Wiki-En dataset) we use the full document to retrieve memory.

**Elements atomic number:** We include a task on predicting the atomic number of different elements. The model completes a prompt in the form of "The atomic number of {...} is," where {...} is replaced with the name of an element (from a total of 118). The model's generation is then processed to extract integer numerical values, and a response is accepted if it matches the actual atomic number. Random baseline performance for this task is 0%. We use the prompt, including the element name, for memory retrieval.

The elements atomic number evaluation is particularly interesting because each query has a specific keyword: the element name. This allows us to count the frequency of each element's name in the dataset (as shown in Fig. 1) to analyze model performance as a function of knowledge scarcity in the pretraining dataset. We note, however, that this analysis has some minor caveats. For example, the word "lead" is both the name of an element and an English verb. In addition, some elements have multiple names; for instance, "Natrium" is an accepted alias for "Sodium". In our frequency calculations, we count all acceptable aliases of each element name.

### B.1 PER DATASET EVALUATION RESULTS

In this section, we provide per-dataset results for all models discussed in Table 1 as shown in Table 11.

| Model | ArcC | ArcE | BOOLQ | COQA | HLSW | LOAI | NQ | OBQA | PIQA | SQUAD | TRIVQA | WINGRD | WINGRDE |
|---|---|---|---|---|---|---|---|---|---|---|---|---|---|
| A1 | 26.5 | 53.3 | 59.7 | 21.3 | 44.3 | 48.1 | 1.9 | 34.0 | 69.3 | 21.2 | 9.3 | 71.4 | 53.4 |
| A2-Generic | 28.2 | 56.0 | 60.0 | 23.3 | 46.6 | 51.4 | 2.2 | 35.8 | 69.0 | 25.2 | 12.1 | 73.3 | 54.3 |
| A2-Fetched | 34.2 | 63.9 | 58.3 | 23.5 | 53.1 | 54.1 | 4.0 | 36.4 | 73.2 | 25.8 | 17.3 | 75.1 | 55.5 |
| B1 | 33.9 | 62.4 | 56.2 | 31.3 | 55.8 | 57.2 | 4.4 | 38.8 | 73.4 | 32.0 | 17.2 | 77.7 | 59.5 |
| B2-generic | 34.0 | 63.6 | 64.2 | 33.6 | 57.6 | 60.8 | 5.2 | 39.2 | 73.1 | 32.8 | 19.5 | 79.9 | 60.6 |
| B2-Fetched | 41.1 | 69.1 | 64.5 | 32.6 | 63.4 | 64.4 | 7.3 | 41.4 | 76.3 | 34.1 | 28.4 | 81.0 | 61.5 |
| C1 | 43.2 | 70.6 | 58.7 | 39.9 | 68.9 | 67.6 | 9.0 | 44.0 | 77.1 | 47.6 | 35.1 | 85.7 | 67.6 |
| C2-Generic | 44.3 | 72.2 | 68.7 | 42.7 | 70.6 | 69.4 | 9.8 | 46.2 | 77.2 | 49.9 | 38.9 | 87.2 | 68.7 |
| C2-Fetched | 48.4 | 75.3 | 70.2 | 42.0 | 73.6 | 70.5 | 12.3 | 49.0 | 80.4 | 49.0 | 45.2 | 87.2 | 68.1 |

Table 11: Per dataset results of all models discussed in Table 1.

## C DATA CLUSTERING DETAILS

We cluster the training set with hierarchical clustering (Grangier et al., 2025). We build a clustering tree: each node in the tree is associated with a cluster centroid. The examples traverse the tree from top to bottom, selecting the node corresponding to the closest centroids among the current node's children.

Before training the clustering tree, we segment our dataset into non-overlapping 2,048 token windows and compute sentence-BERT embedding for every window. We rely on the sentence-BERT `MiniLM-L6-v2` model (Reimers & Gurevych, 2019). This process associates each segment of text with a 384 dimensional vector.

The training of the tree proceed from root to leaves. Iteratively, a new level is built by applying k-means to a subset of the examples belonging to each node. We built a tree of depth up to 4, always splitting nodes in 16 clusters. For k-means, we normalize the Euclidean norm of the vectors prior to clustering. We train the model via Expectation Maximization using k-means++ initialization (Arthur & Vassilvitskii, 2006). At each step, we sample 6,400 new examples. With 20 steps, we visit 128k examples. To ensure a cluster distribution close to uniform, we monitor the cluster sizes at each assignment steps. If a cluster is larger than our balancing limit ($0.094 \simeq 1.5 \times 1/16$), we split evenly at random its assignments with the smallest cluster, as suggested by Jegou et al. (2010).

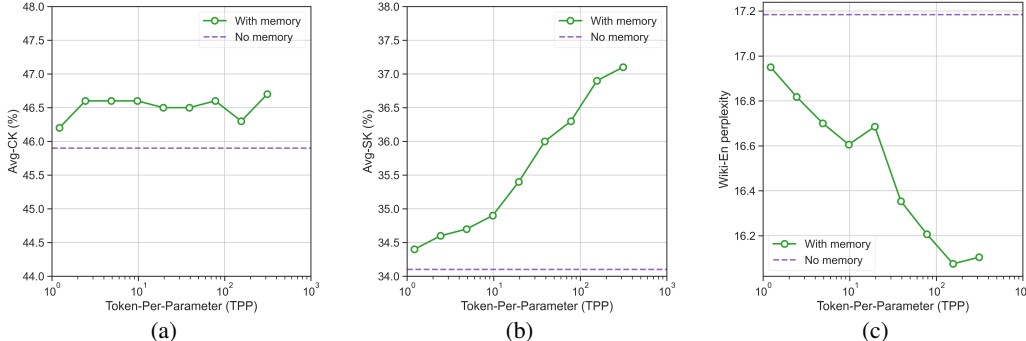

Figure 7: Common-Knowledge (Avg-CK), Specific-Knowledge (Avg-SK), and perplexity on Wiki-En as a function of tokens-per-parameter (TPP). The anchor model is a pretrained 160M model (row A1 in Table 1) and is kept frozen during memory learning.

## D  Tokens per parameter in memory learning

While scaling laws exist for regular language model pretraining that identify the training budget–optimal choice for token-per-parameter (TPP) (Hoffmann et al., 2022), these laws may not hold for memory learning due to its different learning dynamics. Here, we experiment with the effect of TPP when learning memories.

We consider the $(0,0,16,0)$ memory configuration on top of a pretrained 160M anchor model (row A1 in Table 1). This corresponds to 4,096 memories, each with 860,160 parameters, for a total of 3.5B parameters. We freeze the anchor model and train the memories with $2^{32}, 2^{33}, \dots, 2^{40}$ total tokens, corresponding to TPP values ranging from $\simeq 1$ to 312. Results are shown in Fig. 7.

In regular language modeling, knowledge (as in conditional likelihoods) is picked up into parameters when it, and similar content, is repeated often (with many TPP) and receives constructive gradient updates, and forgotten if it is rare and gets destructive gradient updates by dissimilar content (Ghosal et al., 2025). This means that long-tail information is forgotten after it is seen in a batch because the following batches send destructive gradient updates. What lasts in joint parameters is common knowledge, which occurs more often and with more aligned gradients. We demonstrate this effect in Fig. 1.

In the proposed memory learning method, however, memory parameters are shielded in that they are activated and updated only on sequences of a similar topic. This both reduces the times where knowledge can be overwritten and the dissimilarity of possible other gradients. On average, memory parameters corresponding to level $l$ are updated $16^l$ times less often than anchor parameters. For the setup considered above with memory configuration $(0,0,16,0)$, where all memories are at level 3, memory parameters receive one update for every 4,096 sequences in the training set.

Specific knowledge stored in deep memories is shielded from catastrophic forgetting (Toneva et al., 2018), because, unlike regular parameters, it is not overwritten frequently by destructive gradients from unrelated content. Instead, deep memory parameters are only activated when there are constructive gradients of similar content in their clusters, so that they *memorize* this specific knowledge. This behavior is shown in Figs. 7b and 7c, where specific-knowledge benchmark accuracy and wikipedia perplexity steadily improve with increasing TPP. The last point corresponds to a total of 1.1 trillion seen tokens (TPP = 312). Due to computational constraints, we did not scale TPP further.

## E  Fetched memory vs generic memory (additional results)

In Fig. 6a, we showed the performance of the model using fetched and generic memories throughout training, both when anchor parameters are frozen and when they are learnable. A key observation is that when we allow the anchor parameters to be co-trained with the memory bank, we obtain greater improvement compared to the case where the anchor model is frozen (e.g., Avg-SK improves from 39.2% to 40.3% as shown in Table 1). This improvement can be attributed to two factors: 1) when

co-training, the anchor model learns to adapt to the memories more effectively compared to when it is frozen, and 2) the anchor model is exposed to additional tokens, so its performance improves simply due to more training. The latter effect should be minor if the anchor model is already over-trained (i.e., trained with a high TPP) during pretraining and has reached performance saturation, meaning it no longer benefits from additional training.

We also track the gap between the performance of the anchor model using generic memories (a fixed set of memory parameters independent of the input context) and fetched memories throughout co-training, as shown in Fig. 8b and Fig. 8d for the Avg-SK and Wiki-En perplexity metrics, respectively. We observe that the performance gap between fetched and generic memories grows over time (despite using a cosine learning rate schedule), indicating that longer co-training benefits fetched memories more than generic ones. This is expected, as the memory bank has significantly more parameters (4.6B in this example) compared to the anchor model (160M) and a single generic memory (18M), and thus benefits more from extended training.

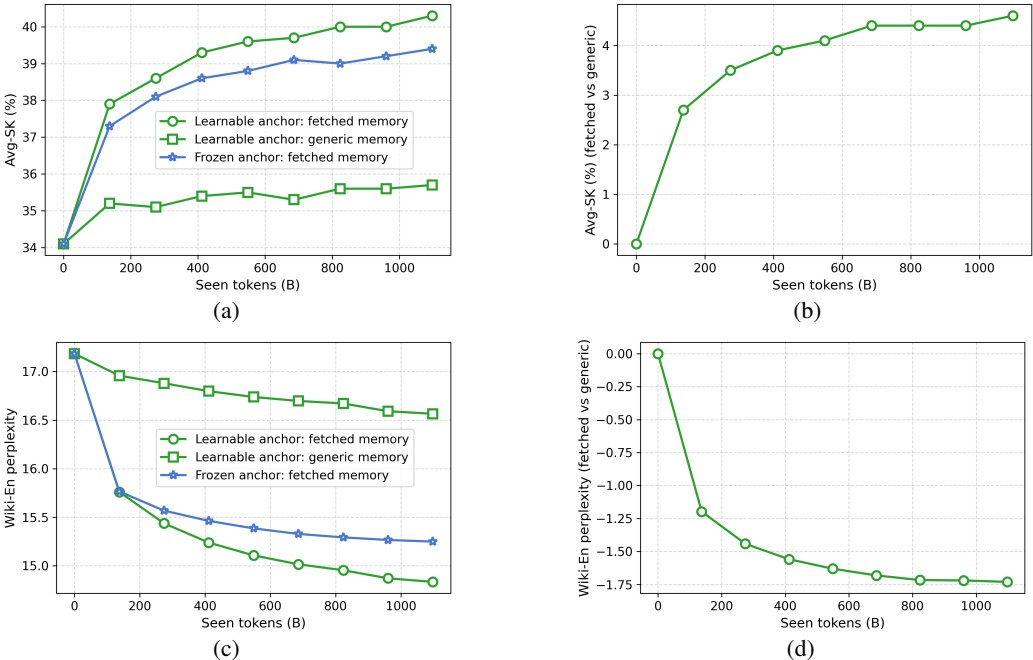

Figure 8: Additional results for the experiment setup of Fig. 6a. Performance improvements from co-training the anchor model and memories, corresponding to row A2 in Table 1. We also show performance when the anchor model is frozen, corresponding to row A3 in Table 1. **(a)** Avg-SK using fetched and generic memories, **(b)** Avg-SK gap between fetched and generic memories, which grows as training progresses, **(c)** Wiki-En perplexity using fetched and generic memories, **(d)** Wiki-En perplexity gap between fetched and generic memories, which widens as training progresses.

## F    MEMORY LOCATION

So far, we have considered memory parameters to be uniformly distributed across the layers of the model. Meng et al. (2022) showed that model knowledge is mainly captured in the middle layers. We use our memory learning setup to further explore this hypothesis by considering different distributions of memory parameters across layers, as shown in Fig. 9.

Starting from a baseline memory with configuration (0,64,0,0), trained on top of a pretrained and frozen 160M model (row A1 in Table 1), corresponding to the level-2 models in Fig. 3d, we study the effect of distributing memory parameters non-uniformly across the model. We consider three setups—early, mid, and late—where the same number of memory parameters as the uniform baseline are applied only to the first, middle, or last 10 layers of the anchor model (see Fig. 9). As

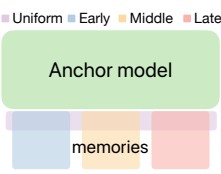

Figure 9: Different distribution of memory along the layers of the anchor model.

| Memory location | Avg-CK (%) ↑ | Avg-SK (%) ↑ | Elements atomic number (%) ↑ | Wiki-En Pplx ↓ |
|---|---|---|---|---|
| No Memory | 45.9 | 34.1 | 1.7 | 17.2 |
| Uniform | 46.7 | **36.8** | 14.4 | **16.0** |
| Early | 45.8 | 34.9 | 2.5 | 16.6 |
| Middle | 46.1 | 35.4 | 1.7 | 16.5 |
| Late | **46.9** | **36.8** | **20.3** | 16.1 |

Table 12: Comparing effectiveness of memories with different distribution over the depth of anchor model, which is kept frozen.

discussed in Appendix A.3, for these experiments the anchor model is frozen, and memories are trained for a total of $2^{38}$ tokens.

Results for each memory placement are shown in Table 12. Consistent with the observations of Lample et al. (2019), using memories in the early layers of the anchor model is less effective than using them uniformly (the default setup) or in the deeper layers.

## G   WHAT TASKS BENEFIT MORE FROM MEMORIES

In Fig. 1, using the atomic number prediction task, we showed that memories can significantly improve performance on tasks requiring long-tail knowledge. To extend this concept to commonly used benchmarks for evaluating pretrained models, we introduce an approximate, quantitative measure of the degree of knowledge specificity for each of the 13 evaluation benchmarks described in Appendix B. For each dataset, we randomly sample 100 entries and prompt GPT-4 (Achiam et al., 2023) to estimate the education level (as a proxy for knowledge specificity) at which a typical person would have acquired the knowledge needed to answer each question. The average of these ratings is used as the knowledge specificity score of the task.

Specifically, we ask the model to rate each question based on the amount of knowledge required, using the following prompt:

> Given the following prompt and answer, what facts should a human know in order to answer the question correctly? What phase of life will a typical person know all required facts? Your response should be in the format of an integer between 0 and 5 based on the following scale:
> 0: Only language understanding, all required information is in the context
> 1: Commonsense facts learned through sensory experiences in childhood
> 2: Facts learned in elementary school
> 3: Facts learned in middle school
> 4: Facts learned in high school
> 5: More specific facts learned later in life
> Respond with only a single integer and nothing else.

In Fig. 10, we plot the accuracy difference between fetched memories and generic memories against the knowledge specificity score for each benchmark, using the 1.4B model trained with memories (row C2 in Table 1). The plot shows a clear positive correlation between knowledge specificity and performance improvement from fetched memories.

We further group the datasets into six common-knowledge (purple) and seven specific-knowledge (green) tasks using a knowledge specificity score threshold of 2.0. On average, fetched memories improve specific-knowledge (Avg-SK) task performance by 3.6 points, while performance on common-knowledge (Avg-CK) tasks remains comparable (64.4 vs. 64.5). Notably, both CK and SK performance show even greater improvement compared to the baseline model (row C1 in Table 1): Avg-CK improves from 61.2% to 64.5%, and Avg-SK improves from 49.7% to 54.9%.

Additionally, we show the same analysis for the 160M (row A2 in Table 1) and 410M (row B2 in Table 1) models trained with memories in Fig. 11 and Fig. 12, respectively.

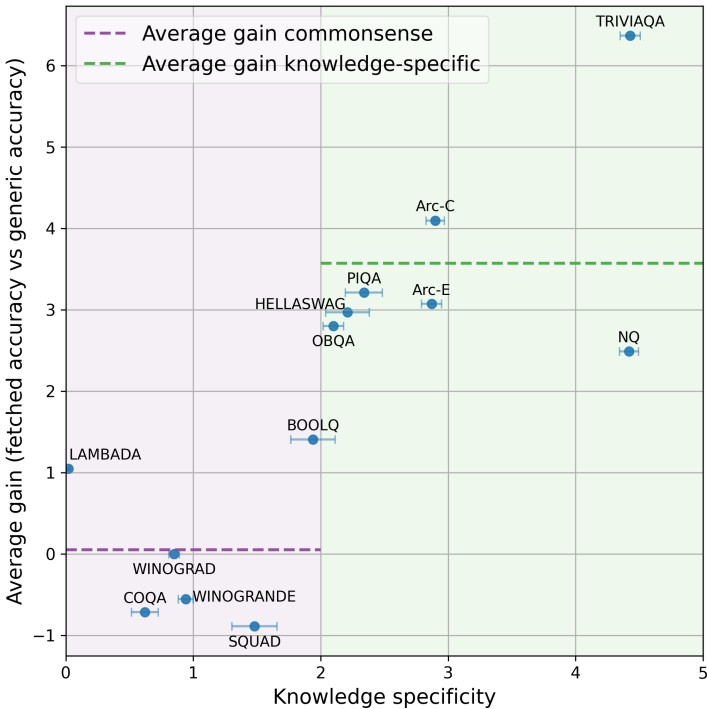

Figure 10: **Fetched memories improve performance on knowledge-intensive benchmarks.** Accuracy gain (fetched memory vs. generic memory) for the 1.4B model (row C2 in Table 1) as a function of the knowledge specificity score of each benchmark. Knowledge specificity is determined by GPT-4 ratings of 100 sampled entries per dataset, and error bars reflect the standard error of the mean. The positive correlation highlights the value of fetched memories for knowledge-intensive tasks. Note that this plot shows the improvement of fetched memories compared to generic memories; the improvement is even greater when comparing fetched memories with the baseline model without memory (row C1 in Table 1).

## H    RETRIEVAL AUGMENTED GENERATION DETAIL

In this section, we provide further details for the experiments in Section 4. We consider two datasets to retrieve documents from: DCLM-Baseline and English Wikipedia 2022, with 6.5 million and 3.2 billion documents, respectively. The average document length in DCLM, using our `EleutherAI/gpt-neox` (Black et al., 2022) tokenizer, is 1,309 tokens (computed from a random 10% subset of the dataset), while for Wikipedia it is 723 tokens. We set the max sequence length at evaluation to 2,560.

As a reference point, we report FLOPs associated with each approach. For RAG models, we use the average sequence length and compute the additional FLOPs on top of a 1024-token context. When using memory, the context length does not increase, but additional memory parameters are fetched and added to the anchor model, increasing runtime FLOPs by $\simeq 10\%$.

For these experiments, we compare augmenting the anchor model with: 1) fetched memory from the learned bank of memories, versus 2) raw documents retrieved from the same dataset the memories were trained on. For high-quality RAG performance, many factors matter, including the instruction-following and long-context capability of the LLM, the retriever's quality, the quality of the datastore, and additional techniques such as self-reflection (Asai et al., 2024).

In this study, we mainly aim to compare learned memories (ours) against contextual memories without additional confounders. We therefore call our retrieval-augmented generations *vanilla* RAG. We use the same retrieval mechanism for both learned memories and RAG: given a query, we identify the context as described in Appendix B. Using the Sentence-BERT (Reimers & Gurevych, 2019)

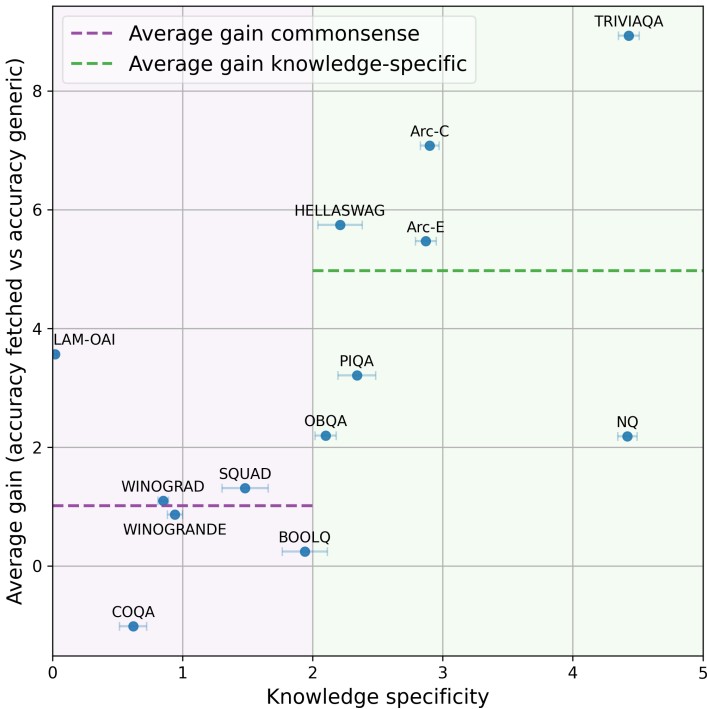

Figure 11: **Fetched memories improve performance on knowledge-intensive benchmarks.** Accuracy gain (fetched memory vs. generic memory) for the 410M model (row B2 in Table 1) as a function of the knowledge specificity score of each benchmark. Knowledge specificity is determined by GPT-4 ratings of 100 sampled entries per dataset, and error bars reflect the standard error of the mean. The positive correlation highlights the value of fetched memories for knowledge-intensive tasks. Note that this plot shows the improvement of fetched memories compared to generic memories; the improvement is even greater when comparing fetched memories with the baseline model without memory (row B1 in Table 1).

`all-MiniLM-L6-v2` embedding model, we first determine the closest level-3 cluster in DCLM (comparing against 4096 centroids). Note that the models with memory in Table 5 are also trained with hierarchical memory configurations up to level 3. We then retrieve the nearest-neighbor (NN) document from within that level 3 cluster (on average $\simeq$ 750k documents, given 3B/4096). This document is then added to the context. To avoid confounders from few-shot complexity in RAG, all tasks are run in a 0-shot setup. We experiment with the 160M, 410M, and 1.4B models, corresponding to rows A1, B1, and C1 in Table 1.

As shown in Table 5, a vanilla NN retrieval from DCLM does not improve baseline model performance for either common-knowledge or specific-knowledge benchmarks. We argue this is mainly due to the low quality of DCLM (a pretraining dataset) when used as a datastore for RAG. Recently, Lyu et al. (2025) showed that with careful filtering, RAG quality can be improved when using web-scale datastores.

To demonstrate the effect of datastore quality here, we also use English Wikipedia to retrieve higher-quality documents for the given context with the same Sentence-BERT model. Results in Table 5 show that RAG-Wiki improves baseline performance on specific-knowledge benchmarks (e.g., from 46.9 to 49.2 for the 1.4B model). However, for common-knowledge benchmarks, RAG-Wiki does not improve over baseline. By contrast, using learned memories (with $\simeq 10\%$ additional parameters relative to baseline) improves performance on both common-knowledge and specific-knowledge benchmarks with lower runtime FLOPs overhead.

Finally, we note that high-quality RAG is complementary to the proposed learned memories and can further enhance performance.

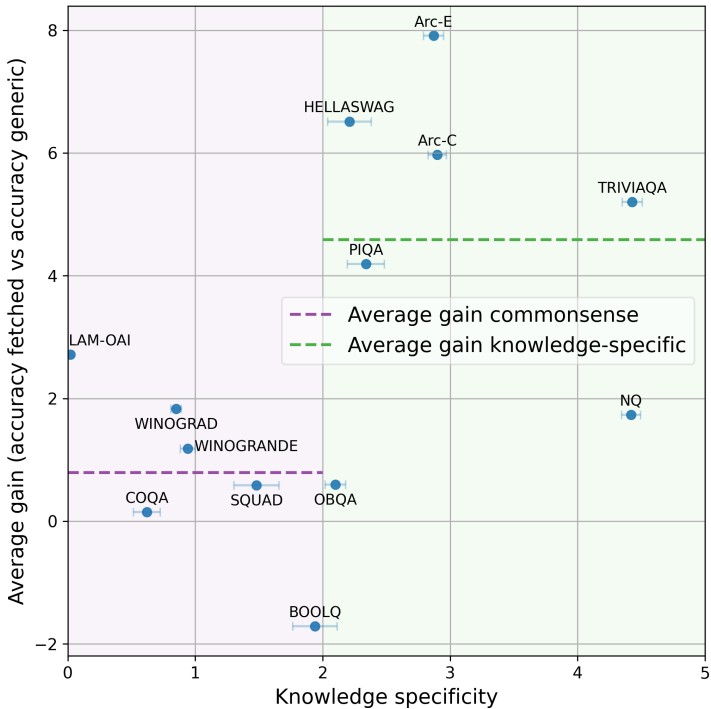

Figure 12: **Fetched memories improve performance on knowledge-intensive benchmarks.** Accuracy gain (fetched memory vs. generic memory) for the 160M model (row A2 in Table 1) as a function of the knowledge specificity score of each benchmark. Knowledge specificity is determined by GPT-4 ratings of 100 sampled entries per dataset, and error bars reflect the standard error of the mean. The positive correlation highlights the value of fetched memories for knowledge-intensive tasks. Note that this plot shows the improvement of fetched memories compared to generic memories; the improvement is even greater when comparing fetched memories with the baseline model without memory (row A1 in Table 1).

## I   MEMORY AUGMENTED TRANSFORMER ARCHITECTURE

In this section, we provide additional detail on different memory augmented transformer architectures that we considered.

**LoRa-Memories:** The transformer block with SwiGLU FFN has seven linear layers, as shown in Fig. 13. We can augment any subset of these with low-rank memories. To avoid exhaustive search, we group the linear layers into three categories based on their role in the transformer block: 1) query and key projection layers, 2) value and output projection layers, and 3) FFN linear layers.

Accordingly, we define three types of LoRa memories: LoRa-QK (shown in Fig. 14), LoRa-OV (shown in Fig. 15), and LoRa-FFN (shown in Fig. 16). Each LoRa consists of two low-rank matrices, $A$ and $B$. The target linear layer $W$ is patched additively as $W + BA$. The size of memories is determined by the rank $r$ of matrices $A$ and $B$. Note that for a model with hidden dimension $d$, inner FFN dimension $d_f$, per attention head dimension $d_h$, $h$ heads, and $l$ layers the size of fetched memory with LoRa type is as follows:

- LoRa-QK memory size: $2rl(d + hd_h)$
- LoRa-OV memory size: $2rl(d + hd_h)$
- LoRa-FFN memory size: $3rl(d + d_f)$

For graceful initialization, so that memories initially have no effect, we initialize $B$ with zeros, as suggested in the original work (Hu et al., 2022), and $A$ with a uniform Kaiming distribution in all

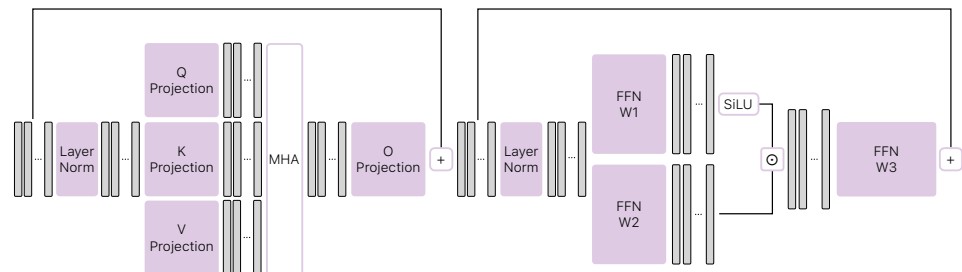

Figure 13: Base architecture of a transformer block with SwiGLU FFN layer.

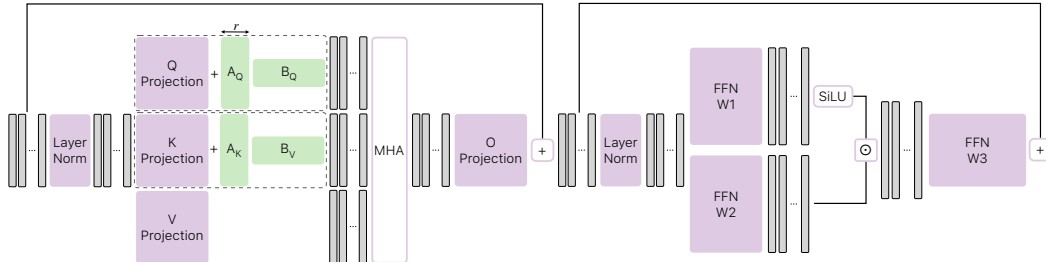

Figure 14: A transformer block with LoRa memories on queries and keys projection layers.

cases. Additionally, we found that a scaling factor of $\alpha = 2$ works best for the LoRa memories considered here.

Aligned with previous observations that knowledge in transformers is primarily stored in FFN layers (Geva et al., 2020; Dai et al., 2022; Yao et al., 2022), we also find that LoRa-FFN mostly outperforms alternative LoRa memories for the same number of learnable parameters, as shown in Fig. 3.

**KV-Memories:** With these memories, we learn additional key and value vectors to augment the input-dependent keys and values. The input-dependent query vectors cross-attend to the learned key and value vectors, and their results are added to the output of multi-head attention. Note that we do not apply causal masking when attending to the learned keys and values. Additionally, we found that KV memories are slightly more effective when used without positional encoding. To ensure no memory effect at initialization, we initialize the learned value vectors with zeros and the learned key vectors with a truncated normal distribution, consistent with other model parameters.

The size of KV memories is determined by the number of key-value vectors ($r$), as shown in Fig. 17, and can be calculated as follows:

- KV memory size: $2rlhd_h$

**FFN-Memories:** for FFN memories, we directly expand the inner dimension of the three linear layers in the SwiGLU FFN as shown in Fig. 18. Similar to other memory types, we initialize memories such that at the beginning of training they have no effect. Therefore, we initialize W1 and W2 FFN memories with truncated normal, and W3 FFN memory with zeros. The size of the FFN memory is determined by their inner dimension and can be calculated as follows:

- FFN memory size: $3rld$

### I.1 MEMORY SIZE CALCULATION

We detail the memory size calculation for one memory configuration, $(256, 64, 16, 0)$ with the 160M anchor model, for additional clarification. As noted in Section 2.1 (last paragraph), a hierarchical configuration $(s_1, s_2, s_3, s_4)$ is given by $(s_1, s_2, s_3, s_4) = c_0 \cdot (r_1, r_2, r_3, r_4)$, where $c_0$ depends on the anchor and memory type, and $r_i$ is the level-$i$ multiplier we choose. For FFN-Memory, $c_0 = 3ld$ (see Fig. 18). For the 160M anchor, for example, $c_0 = 53,760$ (see Table 6).

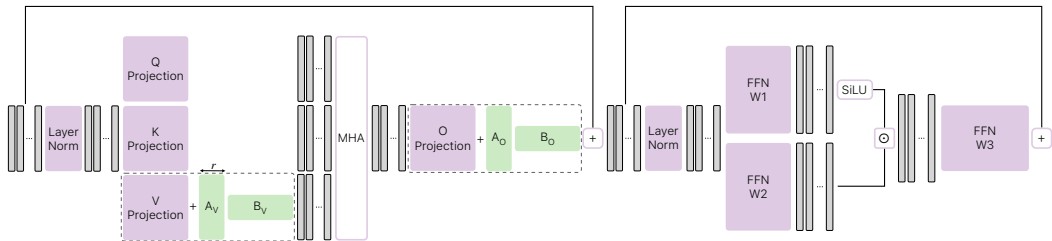

Figure 15: A transformer block with LoRa memories on values and output projection layers.

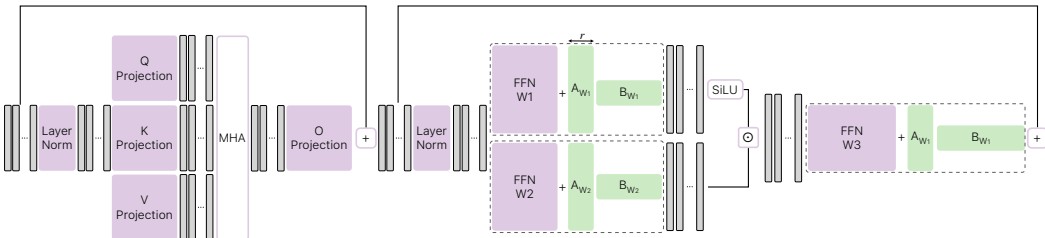

Figure 16: A transformer block with LoRa memories on SwiGLU-FFN linear layers.

With $(r_1, r_2, r_3, r_4) = (256, 64, 16, 0)$:
$s_1 = 53,760 \times 256 = 13,762,560$
$s_2 = 53,760 \times 64 = 3,440,640$
$s_3 = 53,760 \times 16 = 860,160$
$s_4 = 0$.

Fetched memory (per query): $s_1 + s_2 + s_3 + s_4 = 18,063,360$ parameters (~18M). Memory bank (all blocks): $16 s_1 + 16^2 s_2 + 16^3 s_3 + 16^4 s_4 = 4,624,220,160$ parameters (~4.6B).

## J    RETRIEVER FAILURE ANALYSIS

In this section, we provide a few examples of using different fetched memories during generation (max length of 30 tokens) for the 1.4B model with memory (the C2 row in Table 1) under the memory configuration (768, 256, 16, 0). In both examples, we see that as we corrupt the correct memory ID (starting from the finest to the coarsest level), the model's response becomes gradually less accurate.

For the atomic number example, the correct fetched memory ID (8, 130, 2080, 0) not only leads to the correct answer but also adds relevant details, such as the name of the scientist who discovered the element. When we corrupt the memory ID (by replacing it with index 0), the model still gives the correct answer as long as the level-1 memory remains correct, though the additional generated details become wrong. When the level-1 memory is also wrong, the model outputs an incorrect atomic number (92). A similar pattern appears in the example involving scientific names of animals.

```
prompt = "The atomic number of Fermium is" # correct memory (8,130,2080,0)
```

```
model-output-with-memory-id(8,130,2080,0):
" 100.
Fermium is a synthetic element. It was discovered in 1952 by Albert Ghiorso and his team
at the University of California"...
```

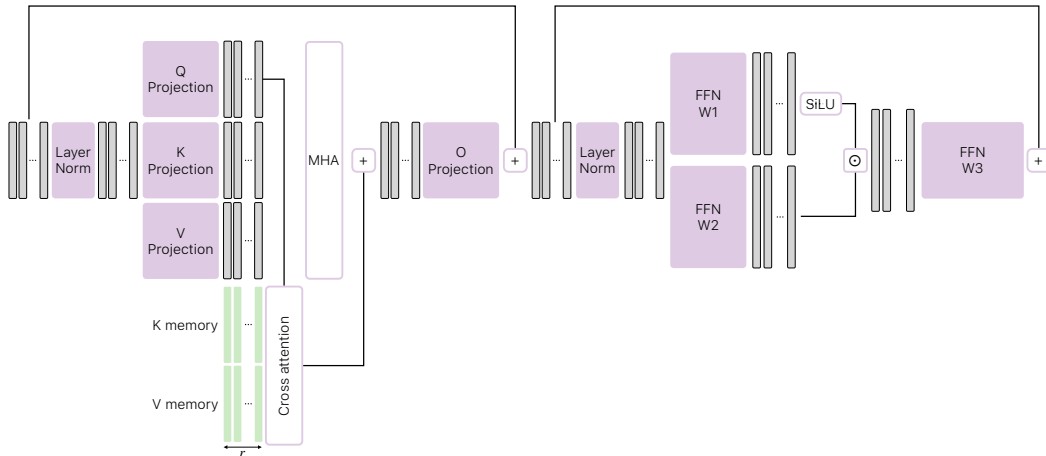

Figure 17: A transformer block with learned KV memories.

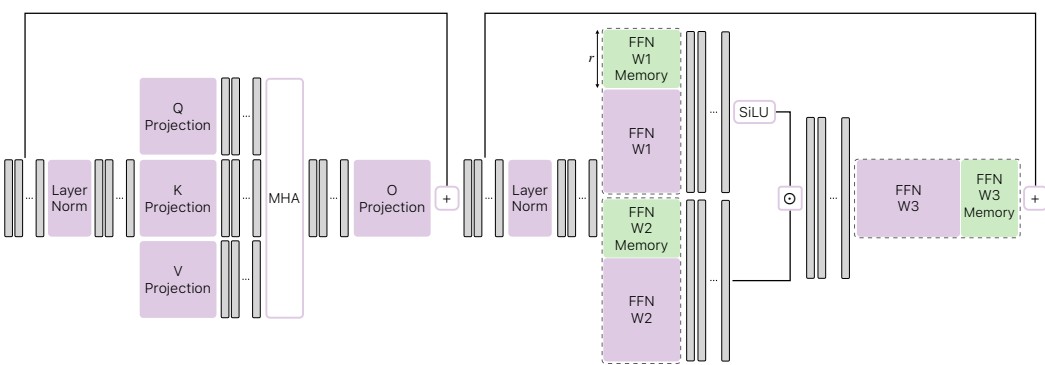

Figure 18: A transformer block with FFN memories.

```
model-output-with-memory-id(8,130,0,0):
" 100.
The atomic weight of Fermium is 262.
The melting point of Fermium is unknown.
The boiling point of"...
```

```
model-output-with-memory-id(8,0,0,0):
" 100.
Fermium is a radioactive metal.
Fermium is a member of the actinide series.
Ferm"...
```

```
model-output-with-memory-id(0,0,0,0):
" 92.
Fermium is a radioactive metal.
Fermium is a radioactive metal.
Fermium is a"...
```

Another example:

> Prompt = "Panthera pardus is the scientific name for" # correct memory
> `(10,168,2703,0)`

> `model-output-with-memory-id(10,168,2703,0):`
> " the leopard. It is a large cat that is native to Africa and Asia. It is the largest of the four big cats. It is also"...

> `model-output-with-memory-id(10,168,0,0):`
> " the leopard.
> The leopard is a large cat, and is the second largest cat in the world.
> The leopard is"

> `model-output-with-memory-id(10,0,0,0):`
> " the lion.
> The lion is a large cat, the largest of the four species of the genus Panthera. Lions are native to Africa"

> `model-output-with-memory-id(0,0,0,0):`
> " the lion.
> The lion is a symbol of royalty and power.
> The lion is a symbol of the sun.
> The lion"

For a more *systematic analysis*, we probed failure modes by stressing the retriever as explained below:

**Setup:** We use a frozen 160M anchor and a non-hierarchical bank `(0,0,0,16)`, so the retriever must pick **one** leaf memory among $16^4 = 65,536$ blocks (that are non-overlapping).

**Corruption protocol:** For each query, we replace the nearest-neighbor leaf with each of its 15 sibling leaves under the same level-3 parent (plausible local misrouting) and average query accuracy over all 16 choices (for each query accuracy is either 0 or 1). Finally we report the mean of this averaged accuracy over the entire benchmark (here considered Arc-Easy and Arc-Challenge with 2376 and 1172 queries, respectively.

| Task | Model | Retriever | Acc. (%) |
|------|-------|-----------|----------|
| ArcE | Baseline | N/A | 55.5 |
| ArcE | w/ mem. | uncorrupted | 60.9 |
| ArcE | w/ mem. | corrupted | 55.8 |
| ArcC | Baseline | N/A | 27.1 |
| ArcC | w/ mem. | uncorrupted | 29.9 |
| ArcC | w/ mem. | corrupted | 28.2 |

Table 13: Failure analysis of the memory retrieval process.

As shown in Table 13, a wrong retrieval reduces the gain but performance remains at or above the no-memory baseline, indicating graceful degradation and robustness to retrieval errors. Further, for the hierarchical case, where two different retrievals can share memory parameters at shallower levels, we expect more robustness.

