# OpenReview forum: "Pretraining with hierarchical memories: separating long-tail and common knowledge"
_ICLR.cc/2026/Conference — ICLR 2026 Poster_

### Official Review · Reviewer_2ftq · 2025-10-31

**Soundness:** 3
**Presentation:** 3
**Contribution:** 3
**Rating:** 4
**Confidence:** 3

**Summary:**

This paper proposes an efficient approach for learning rare knowledge in LLMs than trying to compress them in LLMs parameter. To that, this paper proposes a hierarchical memory bank augmented architecture and a pretraining strategy which demonstrate to achieve similar performance with a small model(roughly half the size of LLM parameters) + very small amount retrieved memory bank parameters. This approach helps with catastrophic forgetting problem, as memory parameters are activated and updated only on sequences of similar topics, thereby reducing susceptibility to forgetting.  The paper discuss 3 approaches of retrieved memory parameter augmentation with the anchor model parameters. There are other related work such as memory layers where FFNs are replaced by learnable memory layers. This work seems a bit incremental although the hierarchical memory back is different.

**Strengths:**

Overall, this paper makes a solid contribution by proposing an approach well-suited for on-device inference. The compact anchor model can reside in fast local memory, while the larger set of knowledge parameters is stored in slower, high-capacity memory. By decoupling knowledge from reasoning capabilities, the method establishes a clear link between training tokens and specific parameter subsets (memories). This design allows targeted selective knowledge update without altering the core anchor model, which can remain publicly accessible.

**Weaknesses:**

This work seems a bit incremental over other memory architecture. There are other parametric memory work, such as memory Layers at scale. It would be good to compare the results with such alternative approaches.

**Questions:**

For the experiments, It would be good to address the following
1. There are other parametric memory work, such as memory Layers at scale. It would be good to compare the results with such alternative approaches.
2. It would be good to share results on how high quality RAG combines with this approach

Other comments:
1. While there is a section discussing adding memory bank on various open models, It would be good to have a discussion regarding if the “learnt” sparse memory bank is transferable, meaning, can be combined with another model with minimal tuning.
2. A more rigorous scaling law analysis such as how does the results scale with increasing number of parameters per memory block
3. How does it compare and combine with MoEs for memorization and reasoning

---

> ### Author Response · Authors · 2025-11-19
> **Thank you for your review. Please see the itemized responses below. [Part 1]**
>
> ## W1 and Q5  (Novelty and position with respect to MoEs):
>
> We appreciate the perspective but disagree that the contribution is merely incremental. Our goal is a pre-training method that forms **parametric memories** with **context-level selection** and a **hierarchical update scheme** aligned with data structure, without changing the backbone or introducing per-token routing. In addition, **we are the first to systematically study which form of memory performs best and how they scale with memory bank size and inference size, both of which are substantial contributions.** Static clustering is used only as auxiliary supervision during pre-training and can be relaxed after training. While exploring a dynamically learned retriever is interesting, it is orthogonal to and outside the scope of this work. In terms of impact, a 160M anchor plus ~10% retrieved memories (<180M total) matches a 410M dense baseline, demonstrating a significant efficiency–accuracy gain. This is not a weak effect and shows the practical value of our design.
>
> Key contributions and distinctions from prior work (MoEs and memory-augmented models):
>
> * **Systematic parametric-memory design**: We are, to our knowledge, the first to **systematically compare** multiple parametric memory forms during pre-training (LoRA variants, learnable KV cache, and new FFN memories) under a unified setup.
> * **Context-based parameter selection (not per-token gating)**: We fetch relevant parameters **once per context**, avoiding the per-token expert routing used in MoEs. This materially reduces memory footprint and bandwidth at inference, which is important for on-device use. In MoEs, most parameters are gated per token (small “anchor”), and active:inactive ratios are ~1:10–1:30; in our method, most parameters remain in the always-used anchor, we add only ~10% memory parameters, and the fetched-to-bank ratio is ~1:300 further improving on-device deployability.
> * **Hierarchical parametric memories aligned to data**: We introduce a **hierarchical memory** whose deeper layers update less frequently but receive gradients from semantically aligned shards induced by nested clustering. This **learning mechanism** is, to our knowledge, unexplored in prior work.
> * **Large-scale efficacy**: Across trillion-token pre-training, **10% retrieved memory** yields performance on par with or better than a **>2$\times$ larger dense model**, showing that the proposed structure delivers nontrivial efficiency gains.
>
> We will clarify these distinctions and empirical findings in the revision and explicitly position dynamic retrievers/routers as promising future work rather than core to our claims.
>
> ---
> ## W1 & Q1 (Position with respect to memory Layers at scale [1])
>
> We have discussed the memory Layers at scale [1] in the Related work section. Similar to MoE approaches this approach is selecting a memory (or an expert in MoE) **per token**, make it vastly different from our proposed method. As long as an on-device use case is concerned, which is the motivation of our work, models that require per-token parameter selection are not suitable since they would still need to have random access to **all parameters** which is not feasible for on-device given limited memory. The key idea in our proposed method is to **fetch memory parameters per context** hence make it suitable for on-device use cases with minimal memory loading overhead.
>
> [1] Berges, Vincent-Pierre, et al. "Memory layers at scale." arXiv preprint arXiv:2412.09764 (2024).

---

> ### Author Response · Authors · 2025-11-19
> **Thank you for your review. Please see the itemized responses below. [Part 2]**
>
> ## Q2 (Comparison with RAG)
>
> Thank you for the comment. **We do not claim superiority over RAG**. Section 4 states that **RAG is complementary** to the proposed parametric memories and can be combined for further gains. **We also include RAG with a high-quality datastore** (English Wikipedia), which improves over the baseline even though the model is a base model (no post-training). Note that **this paper focuses on a pre-training strategy, whereas RAG is a runtime technique; a meaningful and sophisticated RAG setup is only possible for an instruction-following model after post-training**. We include RAG, as explained below, to compare two types of memories, not to show that one is superior.
>
> The main goal of the experiments in Table 3 (Table 5 in the revised paper) is to enable a controlled, mechanism-level study. We keep the retriever (Sentence-BERT) and the datastore (DCLM) fixed so the only changing factor is the memory mechanism: contextual (RAG) vs. parametric (our memories). This setup removes confounders when comparing parametric and contextual memories using the same components. Exploring memory methods, both parametric and contextual (RAG), in post-training pipelines is orthogonal and outside the scope of this work.
>
> To further show the complementarity of contextual (RAG) and parametric (our) memories, **we report results combining the two** when using Wikipedia as the datastore on top of the 410M anchor model (corresponding to rows B1 and B2 in Table 1). For the knowledge-specific tasks, RAG-Wiki improves the baseline by +3.1 points, while the proposed parametric memories improve it by +6.0 points. When combining both RAG and parametric memories, the improvement increases to +7.2 points over the baseline. We also include the raw results of NaturalQuestions and TriviaQA, the most knowledge-specific tasks (as shown in Appendix G, Figures 10–12), to further demonstrate the gains from these two different yet complementary memory augmentation mechanisms: contextual and parametric.
>
> | Model | AVG-SK | NaturalQ | TriviaQA |
> |--------------|-------|-------|-------|
> | Baseline | 38.5 | 4.4 | 12.9 |
> | +RAG Wiki | 41.6 (+3.1) | 11.4 | 23.2 |
> | +Memories | 44.5 (+6.0) | 7.3 | 23.0 |
> | +Memories +RAG Wiki | **45.7** (+7.2) | **12.7** | **28.7** |
>
> ---
>
> ## Q3 (Cross model transferability of memories)
>
> Since the memory parameters are attached to the internal feed-forward layers of the transformer blocks, the learned memories are highly dependent on the anchor model’s architecture, both in size (hidden dimension and depth) and in compatibility with the anchor model’s parameters. As a result, they are not expected to be transferable to very different models.
>
> ---
>
> ## Q4 (Scaling law analysis)
>
> **We have included several scaling experiments** in the current paper including: **scaling memory size** (memory bank and fetched memory each, in isolation) as shown in Fig 4a, **scaling anchor model size** as shown in Table 1, and **scaling token per parameters** as shown in Figure 7 in the Appendix. Deriving a scaling law formula for the proposed setup (anchor model + memory bank) under a given training budget is an interesting research direction, which is out of scope for this work.

---

> > ### Author Response · Authors · 2025-11-26
> >
> > Dear Reviewer,
> >
> > Thank you again for your thoughtful feedback on our submission. We have done our best to address all the points raised in your initial review, and we kindly ask you to reconsider your evaluation of the paper in light of our responses and revisions. If there are any remaining concerns or suggestions regarding our contribution, we would be happy to clarify them.
> >
> > Below is a summary of the changes we made in response to your comments:
> > - Expanded the discussion on the relationship to RAG in Section 4 and reported results combining our proposed memory mechanism with RAG.
> > - Clarified the positioning of our method with respect to MoE based methods in Section 1.
> > - Clarified our contributions at the end of Section 1.
> > - Mentioned deriving a scaling-law formula as a promising direction for future work in Section 6.
> >
> > Thank you very much for your time and consideration.
> >
> > Authors

---

### Official Review · Reviewer_Uqsx · 2025-11-01

**Soundness:** 3
**Presentation:** 4
**Contribution:** 3
**Rating:** 6
**Confidence:** 4

**Summary:**

This paper proposes a new pretraining framework for improved knowledge learning: it separates general and domain-specific knowledge in the pretraining corpus and assigns them to different components of the model. To preserve long-tail knowledge, the authors design a individual hierarchical memory bank that is selectively retrieved and updated when training on similar text corpora, ensuring each submodel in the bank becomes proficient at its assigned tasks. Eperiments show that models ranging in size from 160M to 21B parameters benefit from this memory-bank design while training costs remain moderate.

**Strengths:**

1. Although the hierarchical design is not new, applying it to language model pretraining to preserve long-tail information is innovative.

2. The model-bank design choices — including parameter placement, parameter sizes, and retrieval ratios — are well supported by extensive preliminary experiments.

3. The effectiveness of pretraining within this framework has been demonstrated at scale up to 1.4B-parameter models in the main page, indicating strong scaling potential.

4. Plugging the memory bank post hoc into frozen models has also proven effective, demonstrating the design’s generality; moreover, the memory module can be quickly pre-trained on private or corporate data to meet different needs.

5. The memory bank design enables on-device deployment by loading only the anchor model and retrieved parameters while storing the bulk of knowledge parameters in external storage.

**Weaknesses:**

1. The authors claim that using a base model as an anchor better captures common knowledge and reasoning capabilities, and that augmenting it with a memory bank benefits long-tail knowledge tasks. However, I did not find any evaluation of improvements in the system’s reasoning ability.

2. Although the authors argue that this hierarchical memory bank aligns well with modern computer memory design and therefore offers greater efficiency, they did not provide experiments to support this claim, offering only a brief theoretical analysis at the end of Section 3.

**Questions:**

1. In Figure 3a, is the fetched memory size determined by the memory size $s_2$?

2. “For a fair comparison, during training we use the generic memory with probability 1/(16 + 1) and the fetched memory with probability 16/(16 + 1), where 16 is the clustering division factor. This ensures there is no training bias in favor of the memory bank parameters.” Can you clarify more on the design choice here? Why you choose this approach instead of training a standalone generic memory model?

3. I'm curious how you arrived at a 4.6B memory bank size, given the model bank parameters (256, 64, 16, 0). Can you provide a concrete example?

---

> ### Author Response · Authors · 2025-11-19
> **Thank you for your review. Please see the itemized responses below. [Part 1]**
>
> ## W1 (Improvements in the reasoning ability)
>
> Thank you for your comment. Improving the reasoning capabilities of the anchor model by offloading memorization to memory parameters is mentioned as a supporting motivating factor for separating long-tail knowledge from the main model. We did not pursue this direction in the current work to avoid expanding the scope. Further, we believe this phenomenon would be visible at a larger anchor-model scale, since only at larger scales does long-tail knowledge begin to be learned by the anchor model, and thus offloading it creates opportunities for improvement in other capabilities such as reasoning. In the revision, we will clarify that we do not intend to show anchor-model reasoning improvement when co-trained with memories in the current work.
>
> ---
>
> ## W2 (Inference Cost)
>
> Thank you for raising this. Retrieval in our design happens **once per query/context**, not per token, so it is not a frequent per-step cost. Overheads relative to a dense baseline are:
>
> 1. **Routing (one-time per query)**: Computing the SBERT embedding and nearest-neighbor lookup. With all-MiniLM-L6-v2, this is small compared to LLM inference.
> 2. **Fetch (one-time per query)**: Transferring the selected memory from the bank. The latency depends on the storage tier (e.g., GPU RAM, CPU RAM, NVMe, remote); Figure 5 discusses these trade-offs. Prefetching and caching across turns amortize this cost as shown in Figure 5.
> 3. **Inference (per token)**: Extra FLOPs from the added memories. For example, when fetching about 10% as many memory parameters as anchor parameters, the per-token compute overhead is roughly 10%, but this does not necessarily translate to longer latency.
>
> We further materialize the above discussion, for a specific case where memory bank is stored on the host memory, anchor model on device (GPU) memory, and for every query the relevant memory should is fetched from host to device. Results are shown in the table below.
>
> For the following experiment we give a fixed prompt of length 10 to the model and let it generate 20 and 40 tokens. After 5 wamrup runs, we measure times (in milliseconds) as average over 10 runs. Model is 1.4B with (768, 256, 16, 0) memory configuration corresponding to 153.4 M fetched memories out of a bank with 21.1 B parameters. Model inference is on a single H100 GPU and routing (including the inference of the SBERT model is on CPU). We consider two setups: memory bank residing on device memory and residing on host memory. We add this analysis in the revision.
>
> | Num tokens | Setup        | Bank Location | Routing | Memory fetch | Prefill+Generation | Total time|
> |----------------------|--------------|---------------|--------------|--------------------|-----------------------------|--------|
> | 20                   | No memory    | NA            | NA           | NA                 | 507 ms                      | 507 ms |
> | 20                   | With memory  | GPU Memory    | 7 ms         | 0                  | 534 ms                      | 541 ms |
> | 20                   | With memory  | CPU Memory    | 7 ms         | 23 ms              | 534 ms                      | 564 ms |
> | 40                   | No memory    | NA            | NA           | NA                 | 1017 ms                     | 1017 ms|
> | 40                   | With memory  | GPU Memory    | 7 ms         | 0 ms               | 1040 ms                     | 1047 ms|
> | 40                   | With memory  | CPU Memory    | 7 ms         | 23 ms              | 1040 ms                     | 1070 ms|
>
> ---
>
> ## Q1 (Fetched memory clarification)
>
> Fetched memory size is calculated as the number of parameters that is retrieved from the bank and added to the model. It is a function of anchor model architecture and the memory configuration (memory size at each layer). We provide a detailed example in response to Q3. More details are provided in Appendix I. In the revision we include an example for clarification.

---

> ### Author Response · Authors · 2025-11-19
> **Thank you for your review. Please see the itemized responses below. [Part 2]**
>
> ## Q2 (Generic memory clarification)
>
> Thank you for the question. In the **frozen-anchor** setting (all experiments except those marked “co-train” in Table 1), we do train a **standalone generic-memory** model: a single generic memory on top of a frozen anchor.
>
> The $1/(16+1)$ vs. $16/(16+1)$ sampling is used *only* in the *co-train* setup, where the anchor is updated. We sample the *generic* memory with probability $1/(k+1)$ and a *fetched* memory with probability $k/(k+1)$ (with $k=16$, the clustering division factor). This matches the expected number of updates to the generic memory with those to any individual fetched-memory block, at level 1, avoiding a training bias toward the bank, and is biased toward fetched memory at deeper level.
>
> We do not run two separate trainings (one with the bank, one with the generic memory) because that would misalign the *anchor’s update budget*. The generic memory is small (18.1M for a 160M anchor), while the full bank is large (4.6B). Under a fixed tokens-per-parameter budget, the anchor in the generic-only run would receive $\sim300 \times$ fewer updates. The joint sampling scheme keeps the *same anchor* and ensures comparable anchor updates across generic and fetched memories.
>
> ---
> ## Q3 (Memory bank clarification)
>
> Thank you for the question. We detail the memory-size calculation in Appendix I.
>
> As noted in Section 2.1 (last paragraph), a hierarchical configuration $(s_1,s_2,s_3,s_4)$ is given by $(s_1,s_2,s_3,s_4)=c_0\cdot(r_1,r_2,r_3,r_4)$, where $c_0$ depends on the anchor and memory type, and $r_i$ is the level-$i$ multiplier we choose. For FFN-Memory, $c_0=3ld$ (see Figure 18 in the Appendix for more clarity). For the 160M anchor, $c_0=53,760$ (see Table 6 in the revised draft).
>
> With $(r_1,r_2,r_3,r_4)=(256,64,16,0)$:
>
> $s_1=53,760\times256=13,762,560$
>
> $s_2=53,760\times64=3,440,640$
>
> $s_3=53,760\times16=860,160$
>
> $s_4=0$
>
>
> Fetched memory (per query): $s_1+s_2+s_3+s_4=18,063,360$ parameters ($\sim$18M).
> Memory bank (all blocks): $16s_1+16^2 s_2+16^3 s_3+16^4 s_4=4,624,220,160$ parameters ($\sim$4.6B).
>
> We have added this worked example to Appendix I.1 for clarity in the revised paper.

---

> > ### Author Response · Authors · 2025-11-26
> >
> > Dear Reviewer,
> >
> > Thank you again for your positive feedback on our submission. We have done our best to address all the points raised in your initial review, and we kindly ask you to reconsider your evaluation of the paper in light of our responses and revisions. If there are any remaining concerns or suggestions regarding our contribution, we would be happy to clarify them.
> >
> > Below is a summary of the changes we made in response to your comments:
> > - Clarified in Section 1 that our focus is on memory learning rather than improving general reasoning ability.
> > - Included inference-cost benchmarking and discussion in Section 4, with results reported in Table 3.
> > - Added a concrete example of memory-size calculation in Appendix I.1.
> >
> >  Thank you very much for your time and consideration.
> >
> > Authors

---

### Official Review · Reviewer_cCUe · 2025-11-01

**Soundness:** 3
**Presentation:** 3
**Contribution:** 3
**Rating:** 6
**Confidence:** 3

**Summary:**

This paper proposes a memory-augmented architecture for language models that separates common knowledge (stored in "anchor" parameters) from long-tail knowledge (stored in hierarchical parametric memory banks). During pretraining and inference, the model retrieves context-dependent memory blocks from a large hierarchical memory bank organized via clustering of training data. The authors demonstrate that a 160M parameter anchor model augmented with 18M fetched parameters from a 4.6B memory bank achieves performance comparable to a 2 times larger standard model. The paper includes extensive experiments on memory types, scaling, and deployment considerations.

**Strengths:**

- The paper presents a well-motivated approach to separating common reasoning abilities from long-tail factual knowledge, with clear empirical evidence showing that hierarchical memories particularly benefit knowledge-intensive tasks (e.g., atomic number prediction improving from 1.7% to 67.8% for the 160M model).

- The experimental evaluation is comprehensive, covering multiple model sizes (160M to 1.4B parameters), memory configurations, and architectural variants, with thorough ablation studies on memory types, depths, and sizes that provide actionable insights for practitioners.

**Weaknesses:**

- The comparison with retrieval-augmented generation (RAG) seems somewhat unfair, as the authors use "vanilla RAG" without standard techniques like reranking or filtering, and the improvement over baseline when using high-quality datastores (Wiki-En) is actually comparable to the memory approach on specific-knowledge tasks.

- There's insufficient discussion of failure modes and limitations - for instance, what happens when the clustering assigns dissimilar documents to the same cluster? The paper would benefit from error analysis showing when and why the approach fails.

**Questions:**

Is it possible to get some analysis or visualization of what knowledge is actually stored in memory parameters versus anchor parameters? For example, can you show examples of specific facts that are reliably stored in particular memory blocks?

---

> ### Author Response · Authors · 2025-11-19
> **Thank you for your review. Please see the itemized responses below. [Part 1]**
>
> ## W1 (Comparison with RAG)
>
> Thank you for the comment. **We do not claim superiority over RAG**. Section 4 states that **RAG is complementary** to the proposed parametric memories and can be combined for further gains. **We also include RAG with a high-quality datastore** (English Wikipedia), which improves over the baseline even though the model is a base model (no post-training). Note that **this paper focuses on a pre-training strategy, whereas RAG is a runtime technique; a meaningful and sophisticated RAG setup is only possible for an instruction-following model after post-training**. We include RAG, as explained below, to compare two types of memories, not to show that one is superior.
>
> The main goal of the experiments in Table 3 (Table 5 in the revised paper) is to enable a controlled, mechanism-level study. We keep the retriever (Sentence-BERT) and the datastore (DCLM) fixed so the only changing factor is the memory mechanism: contextual (RAG) vs. parametric (our memories). This setup removes confounders when comparing parametric and contextual memories using the same components. Exploring memory methods, both parametric and contextual (RAG), in post-training pipelines is orthogonal and outside the scope of this work.
>
> To further show the complementarity of contextual (RAG) and parametric (our) memories, **we report results combining the two** when using Wikipedia as the datastore on top of the 410M anchor model (corresponding to rows B1 and B2 in Table 1). For the knowledge-specific tasks, RAG-Wiki improves the baseline by +3.1 points, while the proposed parametric memories improve it by +6.0 points. When combining both RAG and parametric memories, the improvement increases to +7.2 points over the baseline. We also include the raw results of NaturalQuestions and TriviaQA, the most knowledge-specific tasks (as shown in Appendix G, Figures 10–12), to further demonstrate the gains from these two different yet complementary memory augmentation mechanisms: contextual and parametric.
>
> | Model | AVG-SK | NaturalQ | TriviaQA |
> |--------------|-------|-------|-------|
> | Baseline | 38.5 | 4.4 | 12.9 |
> | +RAG Wiki | 41.6 (+3.1) | 11.4 | 23.2 |
> | +Memories | 44.5 (+6.0) | 7.3 | 23.0 |
> | +Memories +RAG Wiki | **45.7** (+7.2) | **12.7** | **28.7** |

---

> ### Author Response · Authors · 2025-11-19
> **Thank you for your review. Please see the itemized responses below. [Part 2]**
>
> ## W2 (Retriever failure analysis)
>
> Below we provide a few examples of using different fetched memories during generation (max length of 30 tokens) for the 1.4B model with memory (the C2 row in Table 1) under the memory configuration $(768, 256, 16, 0)$. In both examples, we see that as we corrupt the correct memory ID (starting from the finest to the coarsest level), the model’s response becomes gradually less accurate.
>
> For the atomic number example, the correct fetched memory ID $(8,130,2080,0)$ not only leads to the correct answer but also adds relevant details, such as the name of the scientist who discovered the element. When we corrupt the memory ID (by replacing it with index 0), the model still gives the correct answer as long as the level-1 memory remains correct, though the additional generated details become wrong. When the level-1 memory is also wrong, the model outputs an incorrect atomic number (92). A similar pattern appears in the example involving scientific names of animals.
>
> ```
> prompt = "The atomic number of Fermium is" # -> correct memory (8, 130, 2080, 0)
> ```
> ```
> model_output_with_memory_id(8, 130, 2080, 0):
> " 100.
>
> Fermium is a synthetic element. It was discovered in 1952 by Albert Ghiorso and his team at the University of California"...
> ```
> ```
> model_output_with_memory_id(8, 130, 0, 0):
> " 100.
>
> The atomic weight of Fermium is 262.
>
> The melting point of Fermium is unknown.
>
> The boiling point of"...
> ```
> ```
> model_output_with_memory_id(8, 0, 0, 0):
> " 100.
>
> Fermium is a radioactive metal.
>
> Fermium is a member of the actinide series.
>
> Ferm"...
> ```
> ```
> model_output_with_memory_id(0, 0, 0, 0):
> " 92.
>
> Fermium is a radioactive metal.
>
> Fermium is a radioactive metal.
>
> Fermium is a"...
> ```
>
> Another example:
>
> ```
> Prompt = "Panthera pardus is the scientific name for" # -> correct memory (10, 168, 2703, 0)
> ```
> ```
> model_output_with_memory_id(10, 168, 2703, 0):
> " the leopard. It is a large cat that is native to Africa and Asia. It is the largest of the four big cats. It is also"...
> ```
> ```
> model_output_with_memory_id(10, 168, 0, 0):
> " the leopard.
>
> The leopard is a large cat, and is the second largest cat in the world.
>
> The leopard is"
> ```
> ```
> model_output_with_memory_id(10, 0, 0, 0):
> " the lion.
>
> The lion is a large cat, the largest of the four species of the genus Panthera. Lions are native to Africa"
> ```
> ```
> model_output_with_memory_id(0, 0, 0, 0):
> " the lion.
>
> The lion is a symbol of royalty and power.
>
> The lion is a symbol of the sun.
>
> The lion"
> ```
>
> For a more **systematic analysis**, we probed failure modes by stressing the retriever as explained below:
>
> **Setup**: We use a frozen 160M anchor and a non-hierarchical bank $(0,0,0,16)$, so the retriever must pick **one** leaf memory among $16^4=65,536$ blocks (that are non-overlapping).
>
> **Corruption protocol**: For each query, we replace the nearest-neighbor leaf with each of its 15 sibling leaves under the same level-3 parent (plausible local misrouting) and average query accuracy over all 16 choices (for each query accuracy is either 0 or 1). Finally we report the mean of this averaged accuracy over the entire benchmark (here considered Arc-Easy and Arc-Challenge with 2376 and  1172 queries, respectively.
>
> | Benchmark | Model | Retriever | Accuracy |
> |-----------------|--------------|-------------|----------|
> | Arc Easy | No Memory | N/A | 55.5 |
> | Arc Easy | With Memory | uncorrupted | 60.9 |
> | Arc Easy | With Memory | corrupted | 55.8 |
> | Arc Challenge | No Memory | N/A | 27.1 |
> | Arc Challenge | With Memory | uncorrupted | 29.9 |
> | Arc Challenge | With Memory | corrupted | 28.2 |
>
> We observe a wrong retrieval reduces the gain but performance remains at or above the no-memory baseline, indicating graceful degradation and robustness to retrieval errors. Further, for the hierarchical case, where two different retrievals can share memory parameters at shallower levels, we expect more robustness. We add this discussion to the final version of the paper.
>
> ---
>
> ## Q1 (Analysis of memory knowledge)
> Thank you for the question. There is a one-to-one correspondence between pretraining documents and memory blocks. In general, we can expect each memory block to capture information from the corresponding documents, but not all of it: only the residual information not already captured by the anchor model (and shallower-level memories). This makes it difficult to determine exactly **what is stored in a memory block**. However, the reverse question, **which block stores a specific fact**, is more approachable. The experiments corresponding to Figure 6b in the paper investigate this reverse question. For the chemistry knowledge involving atomic numbers, we verified that blocking certain memory blocks results in a sharp degradation in performance. This indicates that the information about atomic numbers is stored in those specific blocks. We also refer to examples shown in response to W2 above to see effect of replacing memories in model generation.

---

> > ### Author Response · Authors · 2025-11-26
> >
> > Dear Reviewer,
> >
> > Thank you again for your positive feedback on our submission. We have done our best to address all the points raised in your initial review, and we kindly ask you to reconsider your evaluation of the paper in light of our responses and revisions. If there are any remaining concerns or suggestions regarding our contribution, we would be happy to clarify them.
> >
> > Below is a summary of the changes we made in response to your comments:
> > - Expanded the discussion of the relationship to RAG in Section 4 and reported results that combine our proposed memory mechanism with RAG.
> > - Added a failure-case analysis in Section 4, with extended results in Appendix J, including specific examples that show which block stores a specific fact (by replacing them during inference).
> >
> > Thank you very much for your time and consideration.
> >
> > Authors

---

### Official Review · Reviewer_6eti · 2025-11-01

**Soundness:** 3
**Presentation:** 2
**Contribution:** 3
**Rating:** 6
**Confidence:** 3

**Summary:**

This paper proposes a hierarchical memories pretraining framework that enables small models to achieve large-model-level knowledge coverage. By integrating hierarchical clustering-based memory retrieval, the framework decouples world knowledge from model parameters and separately stores common knowledge and long-tail knowledge. Comprehensive evaluations across 13 benchmarks show that the proposed framework achieves notable improvements in efficient hardware implementation and enhanced data privacy.

**Strengths:**

1. The motivation is clearly presented. The paper designs a hierarchical framework combining an anchor model and a memory bank to separately common knowledge and long-tail knowledge.
2. The proposed method is carefully designed, and its scalability is validated through empirical results on multiple benchmarks.
3. The paper is generally well-organized and readable.

**Weaknesses:**

1. The theoretical contribution is relatively limited. While the paper provides empirical evidence that separating the anchor model and memory bank reduces forgetting and improves training stability, the claim lacks theoretical justification or formal analysis.
2. Per-dataset results are not fully presented. The paper mainly reports averaged results (e.g., Avg-CK and Avg-SK) across 13 benchmarks, which provides a concise overview but may obscure dataset-specific behaviors.
3. Lack of empirical evidence for the claimed knowledge separation. The paper presents a clear motivation that hierarchical memories aim to separate common reasoning abilities from long-tail knowledge. However, semantic clustering alone is insufficient to guarantee that higher levels correspond to common knowledge while deeper levels represent long-tail knowledge.
4. Lack comparison on inference cost. The anchor model+memory bank design may require frequent memory retrieval during each inference task. The authors should provide the average delay cost of their design against baselines.
5. Experiment hardware. The authors should disclose the hardware usage involved in pre-training and inference.
6. Typo. In Figure 2, level 4 should have 64k clusters, instead of 65k.

**Questions:**

1. The figure obscures key lines and does not specify which parameters were fixed. Could the authors clarify what factors were controlled? In addition, some key results in the figure are visually obscured.
2. Is the model performance strongly correlated with the clustering quality of ϕ(x)?
3. How is the number of levels p or division factor k chosen, and how sensitive is performance to these hyperparameters?
4. It is unclear whether the retriever R(x;W) is trained jointly with the anchor model, or whether its cluster assignments are frozen after initialization. How are gradients propagated through R(x;W)?

---

> ### Author Response · Authors · 2025-11-19
> **Thank you for your review. Please see the itemized responses below. [Part 1]**
>
> ## W1 (Theoretical contribution)
> Thank you for the concern. Our contribution is primarily empirical, but it is backed by existing theory (Mem-Sinks [1]). In standard LM training, all parameters update on every step, mixing gradients from dissimilar documents and amplifying interference, which hurts long-tail retention. In our setup, a memory parameter at level l is activated about 16^{-l} as often as the anchor. Thus it receives fewer, more homogeneous updates, which lowers cross-topic gradient covariance and stabilizes training and reduces forgetting of long-tail knowledge, as predicted by the Mem-Sinks framework. We sketch this in lines 280-289 in the revised draft (as well as Appendix D) and will extend the discussion in the camera-ready version.
>
> ---
> ## W2 (Per-dataset results)
> Thank you for the comment. We used averages to keep the main text concise; full per-dataset results are in Appendix G (Figs. 10–12). In the revision, we also add a per-dataset table covering all models in Table 1 as shown below:
>
>
> | Model | ArcC | ArcE | BOOLQ | COQA | HELLASWAG | LAMBADA | NQ | OBQA | PIQA | SQUAD | TRIVIAQA | WINOGRAD | WINOGRANDE |
> |--------------|-------|-------|-------|-------|-------|-------|-------|-------|-------|-------|-------|-------|-------|
> | A1 | 26.5 | 53.3 | 59.7 | 21.3 | 44.3 | 48.1 | 1.9 | 34.0 | 69.3 | 21.2 | 9.3 | 71.4 | 53.4 |
> | A2-Generic | 28.2 | 56.0 | 60.0 | 23.3 | 46.6 | 51.4 | 2.2 | 35.8 | 69.0 | 25.2 | 12.1 | 73.3 | 54.3 |
> | A2-Fetched | 34.2 | 63.9 | 58.3 | 23.5 | 53.1 | 54.1 | 4.0 | 36.4 | 73.2 | 25.8 | 17.3 | 75.1 | 55.5 |
> | B1 | 33.9 | 62.4 | 56.2 | 31.3 | 55.8 | 57.2 | 4.4 | 38.8 | 73.4 | 32.0 | 17.2 | 77.7 | 59.5 |
> | B2-generic | 34.0 | 63.6 | 64.2 | 33.6 | 57.6 | 60.8 | 5.2 | 39.2 | 73.1 | 32.8 | 19.5 | 79.9 | 60.6 |
> | B2-Fetched | 41.1 | 69.1 | 64.5 | 32.6 | 63.4 | 64.4 | 7.3 | 41.4 | 76.3 | 34.1 | 28.4 | 81.0 | 61.5 |
> | C1 | 43.2 | 70.6 | 58.7 | 39.9 | 68.9 | 67.6 | 9.0 | 44.0 | 77.1 | 47.6 | 35.1 | 85.7 | 67.6 |
> | C2-Generic | 44.3 | 72.2 | 68.7 | 42.7 | 70.6 | 69.4 | 9.8 | 46.2 | 77.2 | 49.9 | 38.9 | 87.2 | 68.7 |
> | C2-Fetched | 48.4 | 75.3 | 70.2 | 42.0 | 73.6 | 70.5 | 12.3 | 49.0 | 80.4 | 49.0 | 45.2 | 87.2 | 68.1 |
>
> ---
>
> ## W3 (Empirical evidence for knowledge separation)
>
> We would like to explicitly separate two related contributions in the paper:
>
> 1. **Separating parameters into an anchor and memories** (*not necessarily hierarchical*) allows the anchor to capture common knowledge and the memories to capture long-tail knowledge. Defining and measuring “common” vs. “long-tail” knowledge is non-trivial, so we provide two empirical probes:
>    - **Knowledge-specificity correlation (13 benchmarks):** For each dataset, we sample 100 QA pairs and ask ChatGPT to rate “knowledge specificity” on a 0–5 Likert scale. Gains from memories increase with this score (Appendix G, Figs. 10–12), showing that memories help more for tail knowledge.
>
>    - **Frequency-controlled elements task:** In the elements–atomic-number evaluation, each query targets a unique entity (the element name). We use its pre-training frequency as a proxy for tailness. Figure 1 (right) shows that gains over the no-memory baseline increase as we move into the tail regime. Also, blocking the relevant subset of the memory bank sharply reduces accuracy (Fig. 6b), showing that the needed facts reside in a narrow portion of the bank.
>
> 2. **Hierarchical memories** provide a way to capture a hierarchy of knowledge, from common to more specific. We agree that semantic clustering does not guarantee this behavior. To support the intuition, we refer to one empirical probe in the paper (depth vs. specificity) and add a justification based on training dynamics. We agree that neither of these guarantees learning a true hierarchy, and we will revise the wording in the paper to clarify this.
>
>    - **Depth vs. specificity:** For a fixed runtime memory budget, deeper memories yield larger gains (Fig. 3c), consistent with deeper levels capturing more specific knowledge.
>
>    - **Training dynamics argument:** Let the commonality of a fact be the number of times it appears in the pre-training data. A fact that appears $f$ times can send gradients to at most $f$ memory blocks at each level, *even under random clustering*. With semantic clustering, the number of memory blocks “touched” by a fact is expected to be significantly less than $f$. As a result, most of the $65k$ clusters at level 4 receive no gradient from any single fact. The fraction of clusters touched by a fact increases at higher levels (because there are fewer clusters), and this fraction also grows with fact frequency. Thus, higher-level memories are expected to be dominated by more frequent facts, since they win in *gradient-update competition*.

---

> ### Author Response · Authors · 2025-11-19
> **Thank you for your review. Please see the itemized responses below. [Part 2]**
>
> ## W4 (Inference Cost)
>
> Thank you for raising this. Retrieval in our design happens **once per query/context**, not per token, so it is not a frequent per-step cost. Overheads relative to a dense baseline are:
>
> 1. **Routing (one-time per query)**: Computing the SBERT embedding and nearest-neighbor lookup. With all-MiniLM-L6-v2, this is small compared to LLM inference.
> 2. **Fetch (one-time per query)**: Transferring the selected memory from the bank. The latency depends on the storage tier (e.g., GPU RAM, CPU RAM, NVMe, remote); Figure 5 discusses these trade-offs. Prefetching and caching across turns amortize this cost as shown in Figure 5.
> 3. **Inference (per token)**: Extra FLOPs from the added memories. For example, when fetching about 10% as many memory parameters as anchor parameters, the per-token compute overhead is roughly 10%, but this does not necessarily translate to longer latency.
>
> We further materialize the above discussion, for a specific case where memory bank is stored on the host memory, anchor model on device (GPU) memory, and for every query the relevant memory should is fetched from host to device. Results are shown in the table below.
>
> For the following experiment we give a fixed prompt of length 10 to the model and let it generate 20 and 40 tokens. After 5 wamrup runs, we measure times (in milliseconds) as average over 10 runs. Model is 1.4B with (768, 256, 16, 0) memory configuration corresponding to 153.4 M fetched memories out of a bank with 21.1 B parameters. Model inference is on a single H100 GPU and routing (including the inference of the SBERT model is on CPU). We consider two setups: memory bank residing on device memory and residing on host memory. We add this analysis in the revision.
>
> | Num tokens | Setup        | Bank Location | Routing | Memory fetch | Prefill+Generation | Total time|
> |----------------------|--------------|---------------|--------------|--------------------|-----------------------------|--------|
> | 20                   | No memory    | NA            | NA           | NA                 | 507 ms                      | 507 ms |
> | 20                   | With memory  | GPU Memory    | 7 ms         | 0                  | 534 ms                      | 541 ms |
> | 20                   | With memory  | CPU Memory    | 7 ms         | 23 ms              | 534 ms                      | 564 ms |
> | 40                   | No memory    | NA            | NA           | NA                 | 1017 ms                     | 1017 ms|
> | 40                   | With memory  | GPU Memory    | 7 ms         | 0 ms               | 1040 ms                     | 1047 ms|
> | 40                   | With memory  | CPU Memory    | 7 ms         | 23 ms              | 1040 ms                     | 1070 ms|
>
> ---
>
> ## W5 (Experiment hardware)
> Thank you. We trained and evaluated on NVIDIA H100 GPUs. In the revision, we add a dedicated hardware subsection in Appendix A.
>
> ---
> ## W6 (Typo)
> Thank you for the note. Level-4 has $16^4 = 65,536$ clusters, which we abbreviate as $65k$. We will clarify this computation in the figure caption to avoid confusion.
>
> ---
> ## Q1 (Figure cosmetics)
> Could you please clarify what figure you are referring to? We would be happy to update the figure.

---

> > ### Comment · Reviewer_6eti · 2025-11-24
> > **Q1 (Figure cosmetics)**
> >
> > The key data about level 4 in Figure 3c is obscured.

---

> > > ### Author Response · Authors · 2025-11-24
> > > **Fig 3 is updated.**
> > >
> > > Thanks for the clarification. We have updated the legend location in the revised version of Fig 3c. Please note that for level 4 the curve contains only two points due to the computational cost of using larger memory bank sizes. In the current curve, the second point corresponds to a memory bank with 14.1 billion parameters; adding a third point with $4\times$ more memory, as in the other curves, would require a memory bank of 56.4 billion parameters.

---

> > > > ### Author Response · Authors · 2025-11-26
> > > >
> > > > Dear Reviewer,
> > > >
> > > > Thank you again for your positive feedback on our submission. We have done our best to address all the points raised in your review, and we kindly ask you to reconsider your evaluation of the paper in light of our responses and revisions. If there are any remaining concerns or suggestions regarding our submission, we would be happy to clarify them.
> > > >
> > > > Below is a summary of the changes we made in response to your comments:
> > > > - Added per-dataset results in Appendix B (Table 11) and cross-referenced these from Table 1.
> > > > - Added an ablation study on alternative clustering choices (different embedding models, depths, and division factors) at the end of Section 3, with results presented in Table 2.
> > > > - Clarified how hierarchical memories align with the hierarchical-knowledge intuition and its theoretical basis in Section 3 (and Appendix D).
> > > > - Included inference-cost benchmarking and discussion in Section 4, with results in Table 3.
> > > > - Clarified our hardware choice (NVIDIA H100) in Appendix A.
> > > > - Clarified the abbreviation “65k” in the caption of Figure 2.
> > > > - Updated the Figure 3c legend location.
> > > >
> > > > Thank you very much for your time and consideration.
> > > >
> > > > Authors

---

> > > > > ### Comment · Reviewer_6eti · 2025-11-27
> > > > >
> > > > > Thank you for your reply. I will keep my score.

---

> ### Author Response · Authors · 2025-11-19
> **Thank you for your review. Please see the itemized responses below. [Part 3]**
>
> ## Q2 and Q3 (sensitivity to embedding model and clustering)
>
> Inspired by CRISP [2], our method uses offline **clustering as auxiliary supervision during pretraining**, complementing next-token prediction as the primary supervision.  As discussed in [2], we avoid making too deep clustering (large value for $p$) to keep the quality of nested $k$-means, meaning the probability of two semantically related documents end up being in the same cluster is high. To further demonstrate sensitivity to clustering, we designed another ablation, explained below and included in the revision.
>
> 1. **Effect of the clustering embedding model:** To study this, we replaced the *all-MiniLM-L6-v2* model used in the paper with *gte-small* [3], another embedding model shown in [2] (Table 2, Appendix B) to have higher clustering accuracy. We clustered DCLM-Baseline using *gte-small* into the same 4 levels with dividing factor 16. Using the 160M anchor model (pretrained and frozen), we learned memories with configuration $(64, 16, 4, 1)$, once with *all-MiniLM-L6-v2* clustering and once with *gte-small* clustering. We trained for a total of 275B tokens for this ablation. Results in the table below show very small sensitivity to the embedding model choice; the more accurate *gte-small* performs on par with *all-MiniLM-L6-v2*.
>
> 2. **Effect of clustering hyperparameters (depth $p$ and dividing factor $k$):** Using the same *all-MiniLM-L6-v2* embedding model as in the paper, we clustered DCLM-Baseline with $p = 2$ (depth) and $k = 256$ (dividing factor), which still yields $65k$ leaf clusters but at level 2 instead of level 4. Using the same pretrained and frozen 160M anchor model, we learned memories with configuration $(84, 1)$ and compared them to memories learned with configuration $(64, 16, 4, 1)$ under the original $p = 4$, $k = 16$ clustering. These two memory configurations result in the same fetch memory size (4.6M) and memory bank size (4.7B), making the comparison fair. We trained for 275B tokens. Results in the table below show improvements on knowledge-specific tasks. Note that two-level nested clustering with dividing factor 16 leads to a suboptimal one-level clustering with dividing factor 256, which aligns with our observation. For real-world use cases, the optimal choice of clustering and memory configuration also depends on the underlying hardware hierarchy used to store and fetch the memory bank.
>
> In conclusion, the cluster structure (choices of depth $p$ and dividing factor $k$) appears more important than the choice of embedding model. We hope these additional experiments address the reviewer’s request for further ablations on clustering sensitivity.
>
> | Embedding Model | Cluster configuration | Memory Configuration | Fetch Memory size | Memory Bank size | AVG-CK | AVG-SK |
> |----------------------|------------------------------|----------------------|-------------------|------------------|--------|--------|
> | all-MiniLM-L6-v2 | depth = 4, dividing = 16 | (64, 16, 4, 1) | 4.6 M | 4.7 B | 42.8 | 36.6 |
> | gte-small | depth = 4, dividing = 16 | (64, 16, 4, 1) | 4.6 M | 4.7 B | 42.6 | 36.2 |
> | all-MiniLM-L6-v2 | depth = 2, dividing = 256 | (84, 1) | 4.6 M | 4.7 B | 43.1 | 37.5 |
>
> ---
> ## Q4 (Retriever clarification)
> The retriever model is an off-the-shelf sentence-BERT embedding model ($all-MiniLM-L6-v2*) and is frozen throughout. We never backpropagate grandients through the embedding model.
>
> ---
>
> [1]  Ghosal, Gaurav R., Pratyush Maini, and Aditi Raghunathan. "Memorization sinks: Isolating memorization during llm training." arXiv preprint arXiv:2507.09937 (2025).
>
> [2] Grangier, David, et al. "Task-adaptive pretrained language models via clustered-importance sampling." arXiv preprint arXiv:2410.03735 (2024).
>
> [3] Li, Zehan, et al. "Towards general text embeddings with multi-stage contrastive learning." arXiv preprint arXiv:2308.03281(2023).

---

### Official Review · Reviewer_hJ7H · 2025-11-04

**Soundness:** 2
**Presentation:** 2
**Contribution:** 2
**Rating:** 2
**Confidence:** 5

**Summary:**

This paper proposes a memory augmented architecture to improve the efficiency of language models. The system uses a small "anchor" model for common knowledge and reasoning, which is augmented by a large "hierarchical memory bank" designed to store long tail facts. A clustering based retriever selects a small, context relevant set of memory parameters from this bank to be added to the anchor model during inference. The authors present experiments showing that this method can improve performance on knowledge intensive tasks, particularly for rare facts, and that their 160M parameter model with 18M of fetched memory can perform comparably to a larger, 410M parameter baseline model.

**Strengths:**

The paper is clearly written and the core concept is easy to understand. The use of figures, especially Figure 1 and Figure 2, helps illustrate the architectural idea and its intended effect on long tail knowledge.

The authors were thorough in ablating their own method's design choices. The paper includes a systematic study of different memory types (FFN, LoRa, KV), the impact of memory depth, and the effects of bank size versus fetched memory size. The analysis in Figure 1, showing improved accuracy on rare elements, provides a clear demonstration of the mechanism working as intended for that specific task. The paper also shows the method can be applied post hoc to existing open weight models, which suggests some level of generality.

**Weaknesses:**

The paper's conclusions are undermined by significant weaknesses in its experimental design and unaddressed questions about the core mechanism:

1. Static Retriever: The entire method's effectiveness hinges on a static, offline clustering of the training data using an off the shelf Sentence BERT model. This is a major point of failure. The paper provides no sensitivity analysis on the choice of embedding model or clustering algorithm. If this initial, fixed clustering is suboptimal, the model has no way to adapt, and knowledge separation would presumably fail. The system's performance is completely dependent on this external component.

2. Unsupported Claims on Editing and Privacy: The paper makes strong claims about enabling knowledge editing and privacy. The evidence provided is extremely weak. Figure 6b only shows that blocking memory degrades performance on a specific task. This demonstrates the memory is being used, not that facts can be cleanly removed or edited. It fails to address whether long tail knowledge leaks into the anchor model during co training, which would make any privacy or editing claims invalid.

3. Weak RAG Baseline: The comparison to RAG in Table 3 is not strong. The authors compare their method to a vanilla RAG that retrieves a single document from the low quality DCLM pretraining dataset, which they admit performs poorly. Modern RAG systems use far more sophisticated techniques (multiple documents, reranking, chunking). So, by benchmarking against a weak baseline, the paper's claims of superior efficiency and performance over RAG are not well supported.

4. Limited Novelty: The paper frames itself as a novel solution, but it can also be seen as an incremental combination of existing ideas. The concept of separating parameters for knowledge is central to MoEs, and memory augmented networks which are a long standing field of research. The primary novelty is the hierarchical structure and the clustering based pretraining, which as explained above is a weak and static process.

**Questions:**

The authors would need to address several critical points to make their claims more convincing.

1. Why was a static, offline clustering chosen over a learnable retriever that could co train with the model? can the authors provide any analysis on how sensitive the model's performance is to the quality of the initial clustering or the choice of embedding model?

2. Regarding the privacy and editing claims, can the authors provide direct evidence of knowledge separation?

3. Given that the vanilla RAG baseline is weak, how does this method compare to a more robust, modern RAG implementation using a high quality datastore and standard techniques like multi document retrieval and reranking? Without this, the claims of superiority are difficult to evaluate.

4. When does the memory retriever fail? can the authors show examples of prompts where the retriever fetches the wrong or an irrelevant memory block, and how does the model perform in those scenarios?

---

> ### Author Response · Authors · 2025-11-19
> **Thank you for your review. Please see the itemized responses below. [Part 1]**
>
> ## Q1 and W1 (sensitivity to embedding model and clustering)
>
> Thank you for the feedback. **Our contribution is not to train a memory retriever. Instead, we compare different parametric memory architectures, show how to integrate them into pretraining using clustering-based auxiliary supervision, and study how they scale.** Each of these components is novel. For the retriever, we chose to keep it static so we could answer these questions without the retriever acting as a confounder.
>
> Inspired by CRISP [1], our method uses offline **clustering as auxiliary supervision during pretraining**, complementing next-token prediction as the primary supervision. To further demonstrate sensitivity to clustering, we designed another ablation, explained below and included in the revision.
>
> Finally, replacing static retriever with a learnable one (e.g., MoE-style gating) is an architectural change, not a supervision baseline, and is orthogonal to our contribution of showing that simple offline signals can induce knowledge separation without modifying the backbone. Note that learnable gating in MoE chooses among $\sim$10-30 experts, while here we fetch memory from 65k choices, a much harder task. Hence, we studied several other choices as discussed above, kept the memory retriever frozen, and still showed significant accuracy gains.
>
> **New experiment results:**
>
> 1. **Effect of the clustering embedding model:** To study this, we replaced the *all-MiniLM-L6-v2* model used in the paper with *gte-small* [2], another embedding model shown in [1] (Table 2, Appendix B) to have higher clustering accuracy. We clustered DCLM-Baseline using *gte-small* into the same 4 levels with dividing factor 16. Using the 160M anchor model (pretrained and frozen), we learned memories with configuration $(64, 16, 4, 1)$, once with *all-MiniLM-L6-v2* clustering and once with *gte-small* clustering. We trained for a total of 275B tokens for this ablation. Results in the table below show very small sensitivity to the embedding model choice; the more accurate *gte-small* performs on par with *all-MiniLM-L6-v2*.
>
> 2. **Effect of clustering hyperparameters (depth $p$ and dividing factor $k$):** Using the same *all-MiniLM-L6-v2* embedding model as in the paper, we clustered DCLM-Baseline with $p = 2$ (depth) and $k = 256$ (dividing factor), which still yields $65k$ leaf clusters but at level 2 instead of level 4. Using the same pretrained and frozen 160M anchor model, we learned memories with configuration $(84, 1)$ and compared them to memories learned with configuration $(64, 16, 4, 1)$ under the original $p = 4$, $k = 16$ clustering. These two memory configurations result in the same fetch memory size (4.6M) and memory bank size (4.7B), making the comparison fair. We trained for 275B tokens. Results in the table below show improvements on knowledge-specific tasks. Note that two-level nested clustering with dividing factor 16 leads to a suboptimal one-level clustering with dividing factor 256, which aligns with our observation. For real-world use cases, the optimal choice of clustering and memory configuration also depends on the underlying hardware hierarchy used to store and fetch the memory bank.
>
> In conclusion, the cluster structure (choices of depth $p$ and dividing factor $k$) appears more important than the choice of embedding model. We hope these additional experiments address the reviewer’s request for further ablations on clustering sensitivity.
>
> | Embedding Model | Cluster configuration | Memory Configuration | Fetch Memory size | Memory Bank size | AVG-CK | AVG-SK |
> |----------------------|------------------------------|----------------------|-------------------|------------------|--------|--------|
> | all-MiniLM-L6-v2 | depth = 4, dividing = 16 | (64, 16, 4, 1) | 4.6 M | 4.7 B | 42.8 | 36.6 |
> | gte-small | depth = 4, dividing = 16 | (64, 16, 4, 1) | 4.6 M | 4.7 B | 42.6 | 36.2 |
> | all-MiniLM-L6-v2 | depth = 2, dividing = 256 | (84, 1) | 4.6 M | 4.7 B | 43.1 | 37.5 |
>
> ---
>
> [1] Grangier, David, et al. "Task-adaptive pretrained language models via clustered-importance sampling." arXiv preprint arXiv:2410.03735 (2024).
>
> [2] Li, Zehan, et al. "Towards general text embeddings with multi-stage contrastive learning." arXiv preprint arXiv:2308.03281(2023).

---

> ### Author Response · Authors · 2025-11-19
> **Thank you for your review. Please see the itemized responses below. [Part 2]**
>
> ## Q2 and W2 (claims on knowledge editing)
>
> Thank you for the careful reading. **We do not claim any contribution to knowledge editing**; it is mentioned only as motivation. **We will revise the introduction to remove any language that could be interpreted as a contribution in this area**.
>
> Figure 6b shows controllability: blocking the relevant memory from retrieval search degrades performance on queries whose evidence sits in that memory.
>
> Our privacy claim is narrow: access control at the memory level, not formal privacy. Through clustering, there is a one-to-one mapping between memory parameters and pre-training shards; blocking a memory is equivalent to removing the corresponding documents from pretraining dataset. To prevent leakage during co-training, the anchor cab receive gradients only from public dataset (those without a privacy concern); on private batches we stop gradients to the anchor and update only memory parameters. The “frozen-anchor” experiment in the paper reflects this setup. This setup is consistent with the concurrent FlexOlmo [3] design; we will clarify this connection and adjust wording to emphasize controllable access rather than strong editing or privacy guarantees.
>
> [3] Shi, Weijia, et al. "Flexolmo: Open language models for flexible data use." arXiv preprint arXiv:2507.07024 (2025).
>
> ---
>
> ## Q3 and W3 (Comparison with RAG)
>
> Thank you for the comment. **We do not claim superiority over RAG**. Section 4 states that **RAG is complementary** to the proposed parametric memories and can be combined for further gains. **We also include RAG with a high-quality datastore** (English Wikipedia), which improves over the baseline even though the model is a base model (no post-training). Note that **this paper focuses on a pre-training strategy, whereas RAG is a runtime technique; a meaningful and sophisticated RAG setup is only possible for an instruction-following model after post-training**. We include RAG, as explained below, to compare two types of memories, not to show that one is superior.
>
> The main goal of the experiments in Table 3 (Table 5 in the revised paper) is to enable a controlled, mechanism-level study. We keep the retriever (Sentence-BERT) and the datastore (DCLM) fixed so the only changing factor is the memory mechanism: contextual (RAG) vs. parametric (our memories). This setup removes confounders when comparing parametric and contextual memories using the same components. Exploring memory methods, both parametric and contextual (RAG), in post-training pipelines is orthogonal and outside the scope of this work.
>
> To further show the complementarity of contextual (RAG) and parametric (our) memories, **we report results combining the two** when using Wikipedia as the datastore on top of the 410M anchor model (corresponding to rows B1 and B2 in Table 1). For the knowledge-specific tasks, RAG-Wiki improves the baseline by +3.1 points, while the proposed parametric memories improve it by +6.0 points. When combining both RAG and parametric memories, the improvement increases to +7.2 points over the baseline. We also include the raw results of NaturalQuestions and TriviaQA, the most knowledge-specific tasks (as shown in Appendix G, Figures 10–12), to further demonstrate the gains from these two different yet complementary memory augmentation mechanisms: contextual and parametric.
>
> | Model | AVG-SK | NaturalQ | TriviaQA |
> |--------------|-------|-------|-------|
> | Baseline | 38.5 | 4.4 | 12.9 |
> | +RAG Wiki | 41.6 (+3.1) | 11.4 | 23.2 |
> | +Memories | 44.5 (+6.0) | 7.3 | 23.0 |
> | +Memories +RAG Wiki | **45.7** (+7.2) | **12.7** | **28.7** |

---

> ### Author Response · Authors · 2025-11-19
> **Thank you for your review. Please see the itemized responses below. [Part 3]**
>
> ## W4 (Novelty)
>
> We appreciate the perspective but disagree that the contribution is merely incremental. Our goal is a pre-training method that forms **parametric memories** with **context-level selection** and a **hierarchical update scheme** aligned with data structure, without changing the backbone or introducing per-token routing. In addition, **we are the first to systematically study which form of memory performs best and how they scale with memory bank size and inference size, both of which are substantial contributions.** Static clustering is used only as auxiliary supervision during pre-training and can be relaxed after training. While exploring a dynamically learned retriever is interesting, it is orthogonal to and outside the scope of this work. In terms of impact, a 160M anchor plus ~10% retrieved memories (<180M total) matches a 410M dense baseline, demonstrating a significant efficiency–accuracy gain. This is not a weak effect and shows the practical value of our design.
>
> Key contributions and distinctions from prior work (MoEs and memory-augmented models):
>
> * **Systematic parametric-memory design**: We are, to our knowledge, the first to **systematically compare** multiple parametric memory forms during pre-training (LoRA variants, learnable KV cache, and new FFN memories) under a unified setup.
> * **Context-based parameter selection (not per-token gating)**: We fetch relevant parameters **once per context**, avoiding the per-token expert routing used in MoEs. This materially reduces memory footprint and bandwidth at inference, which is important for on-device use. In MoEs, most parameters are gated per token (small “anchor”), and active:inactive ratios are ~1:10–1:30; in our method, most parameters remain in the always-used anchor, we add only ~10% memory parameters, and the fetched-to-bank ratio is ~1:300 further improving on-device deployability.
> * **Hierarchical parametric memories aligned to data**: We introduce a **hierarchical memory** whose deeper layers update less frequently but receive gradients from semantically aligned shards induced by nested clustering. This **learning mechanism** is, to our knowledge, *unexplored* in prior work.
> * **Large-scale efficacy**: Across trillion-token pre-training, 10% retrieved memory yields performance on par with or better than a $>2\times$ larger dense model, showing that the proposed structure delivers nontrivial efficiency gains.
>
> We will clarify these distinctions and empirical findings in the revision and explicitly position dynamic retrievers/routers as promising future work rather than core to our claims.

---

> ### Author Response · Authors · 2025-11-19
> **Thank you for your review. Please see the itemized responses below. [Part 4]**
>
> ## Q4 (Retriever failure analysis)
>
> Below we provide a few examples of using different fetched memories during generation (max length of 30 tokens) for the 1.4B model with memory (the C2 row in Table 1) under the memory configuration $(768, 256, 16, 0)$. In both examples, we see that as we corrupt the correct memory ID (starting from the finest to the coarsest level), the model’s response becomes gradually less accurate.
>
> For the atomic number example, the correct fetched memory ID $(8,130,2080,0)$ not only leads to the correct answer but also adds relevant details, such as the name of the scientist who discovered the element. When we corrupt the memory ID (by replacing it with index 0), the model still gives the correct answer as long as the level-1 memory remains correct, though the additional generated details become wrong. When the level-1 memory is also wrong, the model outputs an incorrect atomic number (92). A similar pattern appears in the example involving scientific names of animals.
>
> ```
> prompt = "The atomic number of Fermium is" # -> correct memory (8, 130, 2080, 0)
> ```
> ```
> model_output_with_memory_id(8, 130, 2080, 0):
> " 100.
>
> Fermium is a synthetic element. It was discovered in 1952 by Albert Ghiorso and his team at the University of California"...
> ```
> ```
> model_output_with_memory_id(8, 130, 0, 0):
> " 100.
>
> The atomic weight of Fermium is 262.
>
> The melting point of Fermium is unknown.
>
> The boiling point of"...
> ```
> ```
> model_output_with_memory_id(8, 0, 0, 0):
> " 100.
>
> Fermium is a radioactive metal.
>
> Fermium is a member of the actinide series.
>
> Ferm"...
> ```
> ```
> model_output_with_memory_id(0, 0, 0, 0):
> " 92.
>
> Fermium is a radioactive metal.
>
> Fermium is a radioactive metal.
>
> Fermium is a"...
> ```
>
> Another example:
>
> ```
> Prompt = "Panthera pardus is the scientific name for" # -> correct memory (10, 168, 2703, 0)
> ```
> ```
> model_output_with_memory_id(10, 168, 2703, 0):
> " the leopard. It is a large cat that is native to Africa and Asia. It is the largest of the four big cats. It is also"...
> ```
> ```
> model_output_with_memory_id(10, 168, 0, 0):
> " the leopard.
>
> The leopard is a large cat, and is the second largest cat in the world.
>
> The leopard is"
> ```
> ```
> model_output_with_memory_id(10, 0, 0, 0):
> " the lion.
>
> The lion is a large cat, the largest of the four species of the genus Panthera. Lions are native to Africa"
> ```
> ```
> model_output_with_memory_id(0, 0, 0, 0):
> " the lion.
>
> The lion is a symbol of royalty and power.
>
> The lion is a symbol of the sun.
>
> The lion"
> ```
>
> For a more **systematic analysis**, we probed failure modes by stressing the retriever as explained below:
>
> **Setup**: We use a frozen 160M anchor and a non-hierarchical bank $(0,0,0,16)$, so the retriever must pick **one** leaf memory among $16^4=65,536$ blocks (that are non-overlapping).
>
> **Corruption protocol**: For each query, we replace the nearest-neighbor leaf with each of its 15 sibling leaves under the same level-3 parent (plausible local misrouting) and average query accuracy over all 16 choices (for each query accuracy is either 0 or 1). Finally we report the mean of this averaged accuracy over the entire benchmark (here considered Arc-Easy and Arc-Challenge with 2376 and  1172 queries, respectively.
>
> | Benchmark | Model | Retriever | Accuracy |
> |-----------------|--------------|-------------|----------|
> | Arc Easy | No Memory | N/A | 55.5 |
> | Arc Easy | With Memory | uncorrupted | 60.9 |
> | Arc Easy | With Memory | corrupted | 55.8 |
> | Arc Challenge | No Memory | N/A | 27.1 |
> | Arc Challenge | With Memory | uncorrupted | 29.9 |
> | Arc Challenge | With Memory | corrupted | 28.2 |
>
> We observe a wrong retrieval reduces the gain but performance remains at or above the no-memory baseline, indicating graceful degradation and robustness to retrieval errors. Further, for the hierarchical case, where two different retrievals can share memory parameters at shallower levels, we expect more robustness. We add this discussion to the final version of the paper.

---

> > ### Author Response · Authors · 2025-11-26
> >
> > Dear Reviewer,
> >
> > Thank you again for your thoughtful feedback on our submission. We have done our best to address all the points raised in your initial review, and we kindly ask you to reconsider your evaluation of the paper in light of our responses and revisions. If there are any remaining concerns or suggestions regarding our contribution, we would be happy to clarify them.
> >
> > Below is a summary of the changes we made in response to your comments:
> > - Added an ablation study on alternative clustering choices (different embedding models, depths, and division factors) at the end of Section 3, with results reported in Table 2.
> > - Added a failure-case analysis in Section 4, with extended results in Appendix J.
> > - Expanded the discussion of the relationship to RAG in Section 4 and reported results that combine our proposed memory mechanism with RAG.
> > - Clarified that our work does not target knowledge editing or privacy objectives in Section 1.
> > - Clarified the positioning of our method with respect to MoE in Section 1.
> > - Clarified our contributions at the end of Section 1.
> > - Added dynamic retrievers as a promising direction for future work in Section 6.
> >
> > Thank you very much for your time and consideration.
> >
> > Authors

---

### Author Response · Authors · 2025-11-24
**The paper draft has been updated to incorporate all reviewer discussions.**

We thank all reviewers once again for their thoughtful feedback. We have incorporated all requested discussions, additional results, and clarifications into the revised draft (**highlighted in purple**). Below we summarize the changes made in response to the reviews. We kindly invite the reviewers to reconsider their assessments in light of these updates, and we welcome any remaining questions or concerns.

Summary of Revisions
- Added an ablation study on alternative clustering choices (different embedding models, depths, and division factors) at the end of Section 3, with results presented in Table 2.
- Included inference-cost benchmarking and discussion in Section 4, with results in Table 3.
- Added a failure-case analysis in Section 4, with extended results in Appendix J.
- Expanded the discussion on the relationship to RAG in Section 4 and reported results combining our proposed memory mechanism with RAG.
- Added a concrete example of memory-size calculation in Appendix I.1.
- Added per-dataset results in Appendix B (Table 11) and cross-referenced these from Table 1.
- Clarified the positioning of our method with respect to MoE in Section 1.
- Clarified our contributions at the end of Section 1.
- Clarified how hierarchical memories align with the hierarchical-knowledge claim.
- Clarified our hardware choice (NVIDIA H100) in Appendix A.
- Clarified that our focus is on memory learning, rather than improving general reasoning ability, in Section 1.
- Clarified that our work does not target knowledge editing or privacy objectives in Section 1.
- Clarified the abbreviation $16^4$ = 65k in the caption of Figure 2.
- Added dynamic retrievers and scaling-law formula as promising directions for future work in Section 6.

---

### Meta-Review · Area_Chair_LcyS · 2025-12-29

**Summary:**

The authors present a framework to combine a relatively light-weight anchor LLM with learnable memory models that can represent long-tail knowledge. The paper was received with borderline scores (2, 4, 6, 6, 6) by the reviewers after the first round, with reviewers highlighting the clear presentation, the exhaustive ablation studies and usefulness of the general research direction. There also where several concerns voiced by the reviewers (see below). A large portion of those (specifically concerns of negatively-leaning reviewers) has been sufficiently addressed in extensive author responses. The main remaining concern, which is hard to address, is limited novelty, which was mentioned by two reviewers. All in all, I believe the paper would have ended in a positively-leaning borderline state after discussion, after at least one reviewer raising the score.

In this borderline situation, I also took a closer look at the paper, mainly to assess the degree of novelty and contribution in the work. The paper is very empirical in nature, evaluating different system design choices in exhaustive experiments and ablations. Due to this empirical nature, the larger scale experiments, and the very relevant topic, I believe the contribution to be quite valuable. Also, I think the presented paradigm is promising to enable further model scaling on weaker devices. Therefore, I decided to go with an accept recommendation.

**Reviewer Concerns:**

*1) Reliance on offline clustering / sensitivity to clustering.* The authors addressed this well by providing further context and two new experiments, one ablating the choice of embedding generator and another that compares two different clustering hyperparameters. While sensitivity to the latter is certainly a limitation of the method, I would consider this concern well-addressed.

*2) Misleading claims on knowledge editing.* Addressed by the authors via editing the draft.

*3) Weak comparison with RAG.* The authors addressed this by clarifying orthogonality to RAG and supporting this with an additional experiment.

*4) Limited Novelty.* The authors clarified contributions. While it is true that the method is put together from existing parts, the overall contribution lies in system building and in empirical evaluation. As such, and after reading the paper, I believe the novelty to be not outstanding but sufficient.

*5) Only empirical / limited theoretical contribution.* Acknowledged by the authors - not an issue per se, especially in such an applied setting.

*6) Per-dataset results are not fully presented.* Addressed by providing results.

*7) Lack of empirical evidence for the claimed knowledge separation.* The authors provide further clarification and point to evidence in appendix.

*8) Lack comparison on inference cost.* Addressed by clarification and and by providing inference cost comparisons.

*9) Discussion of failure modes and limitations.* Addressed by the authors.

*10) No evaluation of improvements in the system’s reasoning ability.* Declared as out of scope by the authors.

**Reviewer Scores:**

I assume the positively leaning reviewers remain with their score (6, 6, 6) (one also stated as such). Reviewer 2ftq (score 4) questions have been sufficiently answered in my opinion. The novelty concern remains unaddressed. The large set of concerns of Reviewer hJ7H (score 2) have been exhaustively answered by the authors. I would consider them addressed, except for the novelty concern. I think the chance is high that at least one of them would have increased their score.

---

### Decision · Program_Chairs · 2026-01-26

Accept (Poster)